# Asymptotic Analysis of Two-Layer Neural Networks after One Gradient Step under Gaussian Mixtures Data with Structure

**Samet Demir[1], Zafer Doğan[1,2]**
[1]MLIP Research Group, KUIS AI Center, Koç University   [2]Department of EEE, Koç University
{sdemir20,zdogan}@ku.edu.tr

## Abstract

In this work, we study the training and generalization performance of two-layer neural networks (NNs) after one gradient descent step under structured data modeled by Gaussian mixtures. While previous research has extensively analyzed this model under isotropic data assumption, such simplifications overlook the complexities inherent in real-world datasets. Our work addresses this limitation by analyzing two-layer NNs under Gaussian mixture data assumption in the asymptotically proportional limit, where the input dimension, number of hidden neurons, and sample size grow with finite ratios. We characterize the training and generalization errors by leveraging recent advancements in Gaussian universality. Specifically, we prove that a high-order polynomial model performs equivalent to the nonlinear neural networks under certain conditions. The degree of the equivalent model is intricately linked to both the "data spread" and the learning rate employed during one gradient step. Through extensive simulations, we demonstrate the equivalence between the original model and its polynomial counterpart across various regression and classification tasks. Additionally, we explore how different properties of Gaussian mixtures affect learning outcomes. Finally, we illustrate experimental results on Fashion-MNIST classification, indicating that our findings can translate to realistic data.

## 1 Introduction

Understanding how neural networks learn from data and generalize to unseen examples is a fundamental problem in deep learning theory (Goodfellow et al., 2016). While significant progress has been made in analyzing two-layer neural networks (Pennington & Worah, 2017; Ba et al., 2022) under simplified data models, such as isotropic Gaussian inputs, these data models fail to capture the intricate structures present in practical datasets (Ba et al., 2023). Therefore, our research focuses on characterizing the training and generalization performance of two-layer neural networks under more realistic data conditions.

The theoretical analysis of two-layer neural networks has largely focused on the so-called lazy-training regime (Ghorbani et al., 2019), where features experience minimal or no training. One example of this regime is called random feature model (Rahimi & Recht, 2007), where the first layer is randomly initialized and fixed while the second layer is trained. Indeed, the random features have been crucial for exploring phenomena such as "double descent" (Mei & Montanari, 2022) observed in over-parameterized models. However, the random feature model lacks feature learning. Recent studies have begun to fill this gap by analyzing two-layer networks where the first layer is trained with a single gradient descent step (Ba et al., 2022; Damian et al., 2022). This shift significantly enhances our understanding of how feature learning influences generalization performance.

Despite recent advancements, much of the existing work with feature learning mainly relies on overly simplistic data distributions, such as isotropic Gaussian or spherical inputs (Moniri et al., 2024; Dandi et al., 2023a; Cui et al., 2024), and occasionally spiked covariance models (Ba et al., 2023; Mousavi-Hosseini et al., 2023). While these assumptions simplify analysis and provide insights, they overlook the complex nature of real-world data. In practical learning tasks, data is often

better represented as a mixture of distributions (Seddik et al., 2020; Dandi et al., 2023b). Moreover, real-world datasets are typically high-dimensional but often exhibit low intrinsic dimensionality (Facco et al., 2017; Spigler et al., 2020). This gap between theoretical data assumptions and actual data distributions highlights the need for more sophisticated analytical frameworks that capture the mixture nature and low-dimensional structure of real-world datasets.

In this work, we study the performance of a two-layer neural network trained with a single gradient descent step under Gaussian mixture data with covariances including low-dimensional structure. Our data model captures both the mixture nature and intrinsic low-dimensionality in real-world datasets. By leveraging recent advancements in Gaussian universality, we provide a comprehensive characterization of training and generalization errors in the asymptotically proportional limit, where the input dimension, number of hidden neurons, and sample size grow proportionally. Our analysis shows that, under specific conditions, a finite-degree polynomial model—referred to as the "Hermite model"—can achieve equivalent training and generalization performance to that of a nonlinear neural network. We find that the degree of the equivalent polynomial model is closely linked to both "data spread" and the learning rate used during the training of the first layer. We release all our code at https://github.com/KU-MLIP/2-Layer-NNs-with-Gaussian-Mixtures-Data.

Our contributions can be summarized as follows:

- We establish a theoretical framework for characterizing the training and generalization errors of two-layer neural networks under Gaussian mixtures data with covariances featuring additional low-dimensional structures.
- We demonstrate that a finite-degree polynomial model serves as an equivalent performance model, simplifying the analysis of neural networks under Gaussian mixtures data assumption.
- Through extensive simulations, including Fashion-MNIST classification, we validate our findings and highlight the significant impact of the structure of data on learning outcomes.

**Notations** We adopt the standard notations established by Goodfellow et al. (2016) throughout this work unless specified otherwise. The spectral norm of a matrix $\boldsymbol{F}$ is denoted as $\|\boldsymbol{F}\|$. We use the notation $f(\cdot) \asymp g(\cdot)$ to indicate that the functions $f$ and $g$ are of the same order with respect to the parameters $k$, $n$, and $m$. The notation $\mathcal{O}(\cdot)$ represents the big-oh notation in relation to these parameters, and we also define $\tilde{\mathcal{O}}(f(\cdot))$ as shorthand for $\mathcal{O}(f(\cdot) \operatorname{polylog} k)$, effectively allowing us to omit polylogarithmic factors for clarity. Element-wise multiplication is indicated by the symbol $\odot$. Additionally, we denote the conditional expectation of a random variable $X$ given a condition $C$ as $\mathbb{E}[X \mid C]$, while $X_{|C}$ is used to refer to the conditional random variable $X \mid C$. We write $3/4^-$ to denote $3/4 - \epsilon$ for some small $\epsilon > 0$.

## 2 SETTING

We consider a supervised learning setup through a two-layer neural network (NN) defined by

$$\frac{1}{\sqrt{k}} \boldsymbol{w}^T \sigma(\boldsymbol{F} \boldsymbol{x}), \tag{1}$$

where $\boldsymbol{x} \in \mathbb{R}^n$ represents the input vector and $\sigma : \mathbb{R} \to \mathbb{R}$ is the nonlinear activation function. In this framework, $\boldsymbol{F} \in \mathbb{R}^{k \times n}$ and $\boldsymbol{w} \in \mathbb{R}^k$ denote the weights (parameters) of the first and second layers of the NN, respectively. In this study, we focus on the training and generalization performance of this model under a simplified training procedure (one gradient descent step on the first layer) under structured Gaussian mixtures data assumption. Our analysis will be in the proportional asymptotic limit where the number of training data, the input dimension, and the number of features jointly diverge with finite ratios. This regime intuitively represents a scenario in which the width of the network is proportional to the size of the dataset, aligning with common practices in model scaling.

**Data model** We consider data samples drawn from a Gaussian mixture model:

$$\boldsymbol{x} \sim \sum_{j=1}^{\mathcal{C}} \rho_j \mathcal{N}(\boldsymbol{\mu}_j, \boldsymbol{\Sigma}_j), \quad and \quad y := \sigma_* \left( \boldsymbol{\xi}^T \boldsymbol{x}, c \right), \tag{2}$$

where $\mathcal{C} \in \mathbb{Z}^+$ is the number of mixture components and $c$ is a random variable denoting the component assignment for the sample $(\boldsymbol{x}, y)$ with $\mathbb{P}(c = j) = \rho_j$ for $j \in \{1, \ldots, \mathcal{C}\}$. Furthermore,

$\boldsymbol{\mu}_j \in \mathbb{R}^n$ and $\boldsymbol{\Sigma}_j \in \mathbb{R}^{n \times n}$ denote the mean and covariance of $j$-th component, respectively. We further assume that $\boldsymbol{\Sigma}_j$ exhibits a certain low dimensional structure that can be described as finite-rank plus identity (see assumption (A.4) in Section 4). We then define $\boldsymbol{\Sigma} := \mathrm{Cov}(\boldsymbol{x})$ as the covariance matrix of the input vector $\boldsymbol{x}$, and note that its spectral norm $\|\boldsymbol{\Sigma}\|$ will be the measure of *data spread* in our context. Finally, $\sigma_* : \mathbb{R}^2 \to \mathbb{R}$ is an unknown label generation function that can be a nonlinear function of $\boldsymbol{\xi}^T \boldsymbol{x}$ for regression problems or the component index $c$ for classification problems. Note that label $y$ depends only on a single direction $\boldsymbol{\xi}$ (a.k.a single-index target function). This is motivated by the fact that the NN (trained with one gradient descent step) can only learn one direction about the labels (Lemma 1). Extension to multi-index target functions is left to future work.

**Training procedure**  We restrict ourselves to a simplified two-stage training procedure introduced in (Ba et al., 2022; Damian et al., 2022), where we first learn features by taking one gradient descent step on the first-layer parameters and then estimate the second-layer parameters separately. This procedure is summarized as follows:

**i) Gradient descent on the first layer:** Given $\{(\tilde{\boldsymbol{x}}_i, \tilde{y}_i)\}_{i=1}^m$ a set of training samples drawn from (2), we first fix $\boldsymbol{w}$ at the initialization and perform a single gradient descent step on $\boldsymbol{F}$ with respect to squared loss. The gradient update with *learning rate* $\eta > 0$ is given as

$$\hat{\boldsymbol{F}} := \boldsymbol{F} + \eta \boldsymbol{G}, \tag{3}$$

where gradient matrix $\boldsymbol{G}$ is defined as

$$\boldsymbol{G} := \frac{1}{m}\left( \frac{1}{\sqrt{k}}\left( \boldsymbol{w}\tilde{\boldsymbol{y}}^T - \frac{1}{\sqrt{k}}\boldsymbol{w}\boldsymbol{w}^T \sigma(\boldsymbol{F}\tilde{\boldsymbol{X}}^T) \right) \odot \sigma'(\boldsymbol{F}\tilde{\boldsymbol{X}}^T) \right) \tilde{\boldsymbol{X}}, \tag{4}$$

with $\tilde{\boldsymbol{X}} := [\tilde{\boldsymbol{x}}_1, \tilde{\boldsymbol{x}}_2, \ldots, \tilde{\boldsymbol{x}}_m]^T$, and $\tilde{\boldsymbol{y}} := [\tilde{y}_1, \tilde{y}_2, \ldots, \tilde{y}_m]^T$.

**ii) Ridge regression for the second layer:** With the trained first layer $\hat{\boldsymbol{F}}$, we then train the second layer weight vector using a new set of training samples $\{(\boldsymbol{x}_i, y_i)\}_{i=1}^m$ drawn from (2), as follows:

$$\hat{\boldsymbol{w}} := \arg\min_{\boldsymbol{w} \in \mathbb{R}^k} \frac{1}{2m} \sum_{i=1}^m \left( y_i - \frac{1}{\sqrt{k}}\boldsymbol{w}^T \sigma(\hat{\boldsymbol{F}}\boldsymbol{x}_i) \right)^2 + \frac{\lambda}{2}\|\boldsymbol{w}\|^2, \tag{5}$$

where $\lambda \geq 0$ is the regularization constant.

Note that if the ridge regression is performed on the same data, then after one gradient step, $\hat{\boldsymbol{F}}$ will no longer be independent of $\boldsymbol{X}$, which would significantly complicate the analysis. Instead, a new set of training data is used to circumvent this difficulty, following the prior work by Ba et al. (2022).

**Performance metrics**  To evaluate the performance of the two-layer neural network defined in equation (1) and trained on the data model outlined in equation (2), we establish key metrics that quantify both training and generalization errors. After training both layers, we define the training error as:

$$\mathcal{T} := \frac{1}{2m} \sum_{i=1}^m \left( y_i - \frac{1}{\sqrt{k}}\hat{\boldsymbol{w}}^T \sigma(\hat{\boldsymbol{F}}\boldsymbol{x}_i) \right)^2 + \frac{\lambda}{2}\|\hat{\boldsymbol{w}}\|^2, \tag{6}$$

which captures the discrepancy between the predicted outputs and the actual labels for the training dataset, incorporating a regularization term controlled by the parameter $\lambda$ to prevent overfitting. In addition, we assess the generalization error, denoted as:

$$\mathcal{G} := \frac{1}{2}\mathbb{E}_{(\boldsymbol{x}, y)}\left[ \left( y - \frac{1}{\sqrt{k}}\hat{\boldsymbol{w}}^T \sigma(\hat{\boldsymbol{F}}\boldsymbol{x}) \right)^2 \right], \tag{7}$$

which reflects the expected prediction error on unseen data. This metric provides insight into how well the model is likely to perform in practice, beyond the training set.

**Scalings of data spread and learning rate**  Our analysis indicates that the combined scaling of the learning rate $\eta$ and data spread $\|\boldsymbol{\Sigma}\|$ is more critical to our theoretical framework than their individual scalings. To capture this relationship, we introduce a "strength parameter" $\beta \in [0, 1]$ governing the scaling $\eta\|\boldsymbol{\Sigma}\| \asymp n^\beta$. We also define a "weighting parameter" $\alpha \in [0, 1]$ that controls individual scalings: $\|\boldsymbol{\Sigma}\| \asymp n^{\beta(1-\alpha)}$ and $\eta \asymp n^{\beta\alpha}$. This parameter interpolates between two extremes: one where $\eta \asymp n^\beta$ and $\|\boldsymbol{\Sigma}\| \asymp 1$, and another where $\eta \asymp 1$ and $\|\boldsymbol{\Sigma}\| \asymp n^\beta$. This setting covers a wide range of scenarios for generalization performance, which we further illustrate in Appendix D.1.

## 3 RELATED WORK

**Random features —**    The random feature model (RFM) was initially proposed as a computationally efficient approximation to kernel methods (Rahimi & Recht, 2007). RFMs are closely related to the Neural Tangent Kernel (NTK) (Jacot et al., 2018) since both of them provide linear approximations of two-layer neural networks (Ghorbani et al., 2020; 2021). Despite their simplicity, RFMs have proven instrumental in understanding various facets of machine learning, including generalization (Mei & Montanari, 2022), transfer learning (Tripuraneni et al., 2021), out-of-distribution performance (Lee et al., 2023), uncertainty quantification (Clarté et al., 2023), and robustness (Hassani & Javanmard, 2024). Recently, the RFM has garnered renewed interest as a means to investigate the behavior of two-layer neural networks, particularly within the lazy training regime, where the parameters of the network experience minimal changes during training (Pennington & Worah, 2017). Comprehensive asymptotic analyses have been conducted for RFMs (Dhifallah & Lu, 2020; Goldt et al., 2020; Mei & Montanari, 2022; Goldt et al., 2022; Hu & Lu, 2023) and their deep counterparts (Schröder et al., 2023; Zavatone-Veth & Pehlevan, 2023; Bosch et al., 2023), further elucidating their theoretical underpinnings and practical implications.

**Universality —**    One approach to the asymptotic analysis of random feature models (RFMs) involves utilizing equivalent models. Under the assumption of isotropic data, specifically when $x \sim \mathcal{N}(0, I_n)$, it has been demonstrated that the random feature model $w^T \sigma(Fx)$ is equivalent to the following linear model (Goldt et al., 2022; Hu & Lu, 2023):

$$w^T(h_0 \mathbf{1} + h_1 Fx + h_2^* z), \tag{8}$$

where $z \sim \mathcal{N}(0, I_k)$ and $h_0, h_1, h_2^* > 0$ are constants. This equivalence is framed within the concept of "universality" from random matrix theory (Couillet & Liao, 2022), as the random features $\sigma(Fx)$ can be replaced with Gaussian features that share equivalent mean and covariance properties (Hu & Lu, 2023). The universality of random features has since been extended to empirical risk minimization, allowing for broader analyses beyond RFMs (Montanari & Saeed, 2022). While the results by Montanari & Saeed (2022) focused on covariate inputs, Dandi et al. (2023b) recently broadened this framework to encompass inputs distributed as mixtures. Furthermore, Demir & Dogan (2024) extended the universality of random features to Gaussian inputs with spiked covariance, highlighting the significance of structured data in RFM applications.

**Feature learning —**    While the existing literature on random features and universality provides valuable insights, it often overlooks the crucial aspect of feature learning inherent in neural networks. Several studies have explored the dynamics of two-layer networks, particularly in the mean-field regime, which examines training behavior with small learning rates (Mei et al., 2018; Bordelon & Pehlevan, 2024). In this context, we focus on two-layer neural networks where the first layer is trained with a single gradient descent step (Ba et al., 2022), addressing the feature learning deficiencies found in random feature models. Notably, Ba et al. (2022) established the importance of the learning rate in surpassing the performance of the linear model represented in equation (8). Subsequent works by Dandi et al. (2023a), Moniri et al. (2024), and Cui et al. (2024) have further analyzed neural networks after one gradient step through equivalent models, specifically for learning rates $\eta \asymp k^s$ with $s \in [0, 1]$ and isotropic Gaussian inputs. In sharp contrast to these studies, our approach considers Gaussian mixture inputs as defined in equation (2), allowing us to investigate the intriguing effects of data distribution on feature learning. While Ba et al. (2023) and Mousavi-Hosseini et al. (2023) examined Gaussian inputs with spiked covariance, their findings lack equivalent models for precise performance characterization and also lack mixture aspect of our data model (2), highlighting the novelty and significance of our work in this area.

**Gaussian mixtures —**    Most of the works discussed in this section, with the exception of Dandi et al. (2023b), have assumed Gaussian or spherical inputs, which do not adequately capture the mixture nature of class-based problems. Additionally, many of these studies have relied on isotropic covariance, limiting their applicability to simplified scenarios. Recently, there has been a growing interest in analyzing the asymptotic performance of various machine learning problems under the assumption that data is generated from a Gaussian mixture model (Mai & Liao, 2019; Mignacco et al., 2020; Loureiro et al., 2021; Kini & Thrampoulidis, 2021; Refinetti et al., 2021). Notably, Refinetti et al. (2021) compared a two-layer neural network trained in the mean-field regime with

a random feature model using Gaussian mixture inputs in a toy example setting. Furthermore, Loureiro et al. (2021) provided an asymptotic performance characterization for generalized linear models under the Gaussian mixture assumption. In contrast to these studies, our work includes feature learning and introduces straightforward equivalent models for analyzing two-layer neural networks, thereby enhancing the understanding of how data distribution impacts learning dynamics.

## 4 ASSUMPTIONS

(A.1) The number of training samples $m$, input dimension $n$, and number of hidden neurons $k$ jointly diverge with finite ratios, which means $m, n, k \to \infty$ while $n/m, m/k \in \mathbb{R}^+$.

(A.2) $\|\boldsymbol{\Sigma}\| \asymp n^{\beta(1-\alpha)}$ and $\eta \asymp n^{\beta\alpha}$ for $\alpha, \beta \in [0, 1]$. Thus, $\eta\|\boldsymbol{\Sigma}\| \asymp n^{\beta}$.

(A.3) The data is generated according to (2). Furthermore, we let $\boldsymbol{\mu}_c = \mathbf{0}$ and $\mathrm{Tr}(\boldsymbol{\Sigma}_c) = \mathrm{Tr}(\boldsymbol{\Sigma}_{\tilde{c}})$ for all $c, \tilde{c} \in \{1, \ldots, \mathcal{C}\}$ to simplify our analysis.

(A.4) Furthermore, $\boldsymbol{\Sigma}_c$ admits the following decomposition for all $c \in \{1, \ldots, \mathcal{C}\}$,

$$\boldsymbol{\Sigma}_c = \boldsymbol{I}_n + \sum_{i=1}^{d_c} \theta_{c,i} \boldsymbol{\gamma}_{c,i} \boldsymbol{\gamma}_{c,i}^T, \tag{9}$$

where $d_c \in \mathbb{Z}^+$, $\theta_{c,i} > 0$ for all $i \in \{1, \ldots, d_c\}$, and $\{\boldsymbol{\gamma}_{c,i}\}_{i=1}^{d_c}$ is a set of orthonormal vectors in $\mathbb{R}^n$. Note that $\max_{c,i} \theta_{c,i} \asymp n^{\beta(1-\alpha)}$ by (A.2).

(A.5) Let $\|\boldsymbol{\xi}\| = C/\|\boldsymbol{\Sigma}^{1/2}\|$ for some $C > 0$ so that $\mathbb{E}[(\boldsymbol{\xi}^T \boldsymbol{x})^2] = \tilde{\mathcal{O}}(1)$.

(A.6) Let $\boldsymbol{F} := [\boldsymbol{f}_1, \boldsymbol{f}_2, \ldots, \boldsymbol{f}_k]^T$ with $\boldsymbol{f}_i \sim \mathcal{N}(0, \boldsymbol{I}_n/\mathrm{Tr}(\boldsymbol{\Sigma}))$, and $\boldsymbol{w} \sim \mathcal{N}(0, \boldsymbol{I}_k/k)$.

(A.7) The target function $\sigma_* : \mathbb{R}^2 \to \mathbb{R}$ is a Lipschitz function.

(A.8) The activation $\sigma : \mathbb{R} \to \mathbb{R}$ is a function with bounded derivatives and it satisfies $\mathbb{E}_{z \sim \mathcal{N}(0,1)}[\sigma(bz)^2] < \infty$, which allows the following Hermite expansion

$$\sigma(x) = \sum_{j=0}^{\infty} \frac{1}{j!} h_j H_j(x/b), \tag{10}$$

where $H_j : \mathbb{R} \to \mathbb{R}$ denotes $j$-th probabilist's Hermite polynomial (O'Donnell, 2014, Chapter 11.2), $h_j := \mathbb{E}_{z \sim \mathcal{N}(0,1)}[H_j(z)\sigma(bz)]$ and $b := \sqrt{n/\mathrm{Tr}(\boldsymbol{\Sigma})} \in \mathbb{R}^+$ by (A.4).

**Discussion of Assumptions** In this work, we adopt several key assumptions to facilitate our analysis, starting with (A.1), which defines the proportional asymptotic limit (or linear scaling regime). This assumption, commonly used in the literature (Hu & Lu, 2023), allows us to interchangeably use parameters $n$, $m$, and $k$ in our derivations. While we recognize the potential for extending to a polynomial scaling regime (Hu et al., 2024), we leave that exploration for future work. Assumptions (A.2) through (A.5) specifically address our Gaussian mixture data model in equation (2). Assumption (A.2) outlines the necessary range for the strength parameter $\beta$, with potential for valuable insights from an extended range. Assumption (A.3) is used to streamline the derivation of Theorem 4. Yet, the zero-mean assumption $\boldsymbol{\mu}_c = \mathbf{0}$ for the mixture components can be relaxed as discussed in Appendix F. Additionally, (A.4) extends the spiked covariance model (Johnstone, 2001; Baik et al., 2005; Ba et al., 2023) by positing a finite-rank plus identity covariance model, inspired by the low intrinsic dimensions of real-world data (Facco et al., 2017; Spigler et al., 2020). Assumption (A.5) is included to prevent diverging labels. Furthermore, (A.6) relates to standard initialization practices for neural network parameters, ensuring $\mathbb{E}[(\boldsymbol{f}_i^T \boldsymbol{x})^2] = 1$ and $\mathbb{E}[\|\boldsymbol{w}\|^2] = 1$. Assumption (A.7) reflects typical expectations regarding labeling functions, while assumption (A.8) pertains to the activation function used in our proofs. Although some functions like polynomials may not meet the bounded derivatives criterion directly, our results remain valid as long as the derivatives are bounded with high probability for inputs of the form $bz$ where $z \sim \mathcal{N}(0, 1)$. Thus, the equivalent polynomial activation in Theorem 4 is encompassed by our activation function assumption.

## 5 MAIN RESULTS

In this section, we present our main results that enhance our understanding of two-layer neural networks under structured data. We start by analyzing the gradient $G$ defined in equation (4), deriving a decomposition in Lemma 1. Then, we decompose $\hat{F}x$, revealing its "structure" and "bulk" components (Lemma 2). This decomposition helps identify a conditional feature map equivalent to $\sigma(\hat{F}x)$ in terms of training and generalization (Theorem 3). To simplify the conditional feature map, we approximate it using a polynomial function, again leveraging the structure-bulk composition of $\hat{F}x$. Specifically, in Theorem 4, we show that the neural network is equivalent to a polynomial model with regards to the training and generalization errors. The equivalent polynomial model facilitates the analysis of the nonlinear activation function through a reduced set of coefficients.

First of all, we consider the gradient matrix $G$ and its decomposition into spike and bulk components in the following lemma, which characterizes the decomposition.

**Lemma 1** (Spike+bulk decomposition of the gradient). *Consider the gradient $G$ defined in (4). It admits the following decomposition*

$$G = uv^T + \Delta, \tag{11}$$

*where $u := \tilde{h}_1 w$ and $v := \tilde{X}^T \tilde{y}/(m\sqrt{k})$, where $\tilde{h}_1 := \mathbb{E}_{z \sim \mathcal{N}(0,1)}[\sigma'(z)]$. Also, $\|u\| = \tilde{\mathcal{O}}(1)$, $\|v\| = \tilde{\mathcal{O}}(k^{-t/2})$ and $\|\Delta\| = \tilde{\mathcal{O}}(k^{-t})$ with high probability, where $t := 1 - \beta(1 - \alpha) \geq 0$.*

*Proof.* Appendix A. □

This lemma is pivotal as it provides a decomposition of the gradient $G$ into a dominant rank-one term $uv^T$ and a negligible residual term $\Delta$. In this context, $u = \tilde{h}_1 w$ represents the scaled second-layer weights, while $v = \frac{1}{m\sqrt{k}} \tilde{X}^T \tilde{y}$ captures the covariance between the inputs and labels. The residual term $\Delta$ exhibits a magnitude of $\mathcal{O}(k^{-t})$, indicating its diminishing contribution relative to the dominant spike term $uv^T$, which has a norm of $\tilde{\mathcal{O}}(k^{-t/2})$. Consequently, the spike term $uv^T$ emerges as the primary component driving the updates to the first-layer weights. This allows us to express the updated feature matrix as $\hat{F} = F + \eta\Delta + \eta uv^T$. With this formulation in hand, we can proceed to analyze $\hat{F}x$ in the subsequent lemma, leveraging the insights gained from the gradient decomposition to further extend our understanding of how $\hat{F}$ and $x$ interact.

**Lemma 2** (Structure+bulk decomposition of $\hat{F}x$). *Suppose that $x$ is conditioned on $c$-th Gaussian component of the mixture (2): $x_{|c} \sim \mathcal{N}(0, \Sigma_c)$. Thus, $\hat{F}x_{|c}$ can be equivalently written as $\hat{F}\Sigma_c^{1/2}z$ for $z = \Sigma_c^{-1/2}x_{|c} \sim \mathcal{N}(0, I_n)$. Then, we use the orthogonal decomposition: $z = \Gamma_c \kappa_c + z^\perp$, where $\Gamma_c := [v, \gamma_{c,1}, \gamma_{c,2}, \ldots, \gamma_{c,d_c}]$ for a set of vectors $\{\gamma_{c,i}\}_{i=1}^{d_c}$ defined in assumption (A.4), $\kappa_c := (\Gamma_c^T \Gamma_c)^{-1}\Gamma_c^T z$ and $z^\perp := (I_n - \Gamma_c(\Gamma_c^T \Gamma_c)^{-1}\Gamma_c^T)z$. This leads to*

$$\hat{F}\Sigma_c^{1/2}z = \underbrace{(F + \eta\Delta)z^\perp}_{F^\perp z^\perp \text{ (Bulk)}} + \underbrace{\hat{F}\Sigma_c^{1/2}\Gamma_c \kappa_c}_{a_{|\kappa_c} \text{ (Structure)}}, \tag{12}$$

*where $F^\perp := F + \eta\Delta$ and $a_{|\kappa_c} := \hat{F}\Sigma_c^{1/2}\Gamma_c \kappa_c$.*

*Proof.* The result directly follows from the definitions, assumption (A.4) and the orthogonality. □

Intuitively, Lemma 2 reveals that $\hat{F}x$ behaves like noise with a mean for a given pair $(c, \kappa_c)$. This insight enables us to interpret $\sigma(\hat{F}x)$ as random features associated with the specific conditions of $(c, \kappa_c)$. By leveraging this observation in conjunction with established universality results for random features (Hu & Lu, 2023; Montanari & Saeed, 2022; Dandi et al., 2023b), we derive the following theorem. This theorem presents a conditional feature map that is equivalent to $\sigma(\hat{F}x)$ in terms of both training and generalization performance. This equivalence not only underscores the relevance of structured data in feature learning but also enriches our theoretical framework by connecting the behavior of neural networks to the well-studied properties of random feature models. Through this approach, we enhance our understanding of how the underlying data distribution influences learning dynamics, paving the way for more nuanced analyses in subsequent sections.

**Theorem 3** (Conditional Gaussian equivalence). *Under the assumptions (A.1)-(A.8), consider feature map $\phi(\boldsymbol{x}) := \sigma(\hat{\boldsymbol{F}}\boldsymbol{x})$ with the definitions in Lemma 2. Then, define the following conditional feature map (conditioned on $c, \boldsymbol{\kappa}_c$)*

$$\hat{\phi}(\boldsymbol{x}; c, \boldsymbol{\kappa}_c) := \boldsymbol{\nu}(c, \boldsymbol{\kappa}_c) + \boldsymbol{\Psi}(c, \boldsymbol{\kappa}_c)\boldsymbol{z}^{\perp} + \boldsymbol{\Phi}(c, \boldsymbol{\kappa}_c)^{1/2}\boldsymbol{g}, \tag{13}$$

*where* $\quad \boldsymbol{\nu}(c, \boldsymbol{\kappa}_c) := \mathbb{E}\left[\sigma(\hat{\boldsymbol{F}}\boldsymbol{x}) \mid c, \boldsymbol{\kappa}_c\right], \qquad \boldsymbol{\Psi}(c, \boldsymbol{\kappa}_c) := \mathbb{E}\left[\sigma(\hat{\boldsymbol{F}}\boldsymbol{x})(\boldsymbol{z}^{\perp})^T \mid c, \boldsymbol{\kappa}_c\right], \quad$ (14)

$$\boldsymbol{\Phi}(c, \boldsymbol{\kappa}_c) := Cov\left(\sigma(\hat{\boldsymbol{F}}\boldsymbol{x}) \mid c, \boldsymbol{\kappa}_c\right) - \boldsymbol{\Psi}(c, \boldsymbol{\kappa}_c)\boldsymbol{\Psi}(c, \boldsymbol{\kappa}_c)^T, \qquad \boldsymbol{g} \sim \mathcal{N}(0, \boldsymbol{I}_n). \tag{15}$$

*Consider replacing the feature map $\phi(\boldsymbol{x})$ with the conditional feature map $\hat{\phi}(\boldsymbol{x}; c, \boldsymbol{\kappa}_c)$. Then,*

  *(i) the training error $\mathcal{T}$ with the feature map $\phi(\boldsymbol{x})$, and that with the conditional feature map $\hat{\phi}(\boldsymbol{x}; c, \boldsymbol{\kappa}_c)$, both converge in probability to the same value,*

  *(ii) the corresponding generalization errors $\mathcal{G}$ also converge in probability to the same value if an additional assumption (A.9) provided in Appendix B hold.*

*Proof.* Appendix B. □

Recall that $c$ denotes the index of the Gaussian component in the input mixture, while $\boldsymbol{\kappa}_c$ represents the alignment with the subspace of the structure defined in Lemma 2. Theorem 3 establishes that, after conditioning on $(c, \boldsymbol{\kappa}_c)$, the feature map $\phi(\boldsymbol{x}) = \sigma(\hat{\boldsymbol{F}}\boldsymbol{x})$ can be effectively substituted with the conditional feature map $\hat{\phi}(\boldsymbol{x}; c, \boldsymbol{\kappa}_c)$ without impacting training and generalization errors. This substitution streamlines our analysis by allowing the feature map to be expressed in terms of conditional expectations and covariances. Theorem 3 sets itself apart from prior results on Gaussian equivalence through its unique application of conditioning. While our findings build on the work of Hu & Lu (2023) and Dandi et al. (2023b), their results do not include similar conditioning due to the absence of the structure described in Lemma 2. Moreover, the conditional Gaussian equivalence from Dandi et al. (2023a) is limited to conditioning on a spike in the gradient under isotropic Gaussian data. In contrast, our result incorporates conditioning on both the mixture component and the structure in Lemma 2, highlighting a more nuanced interplay between data characteristics and feature learning dynamics. While Theorem 3 is compelling on its own, we can further approximate the conditional feature map using a polynomial feature map. To do so, we consider the $i$-th element of $\hat{\boldsymbol{F}}\boldsymbol{x}$ with the decomposition given in Lemma 2, which leads to $a_{i|\kappa_c} + (\boldsymbol{f}_i^{\perp})^T\boldsymbol{z}^{\perp}$ where $a_{i|\kappa_c} := \hat{\boldsymbol{f}}_i^T\boldsymbol{\Sigma}_c^{1/2}\boldsymbol{\Gamma}_c\boldsymbol{\kappa}_c$ and $\boldsymbol{f}_i^{\perp} := \boldsymbol{f}_i + \eta\boldsymbol{\Delta}_i$. For $a_{i|\kappa_c}$, Lemma 8 asserts that if $\frac{l-2}{l-1} < \beta < \frac{l-1}{l}$, then $|a_{i|\kappa_c}|^l = \tilde{\mathcal{O}}(1/k^{1+\epsilon})$ for some $\epsilon > 0$ with high probability. The vanishing nature of $|a_{i|\kappa_c}|$ enables us to approximate $\sigma(a_{i|\kappa_c} + (\boldsymbol{f}_i^{\perp})^T\boldsymbol{z}^{\perp})$ with $\hat{\sigma}_l(a_{i|\kappa_c} + (\boldsymbol{f}_i^{\perp})^T\boldsymbol{z}^{\perp})$, where $\hat{\sigma}_l : \mathbb{R} \to \mathbb{R}$ is a polynomial activation function defined in equation (16). By utilizing this approximation, we can derive the following theorem, which further elucidates the relationship between conditional feature maps and polynomial approximations within our framework.

**Theorem 4.** *Under the assumptions (A.1)-(A.8), let $\sigma$ be an activation function. Suppose that there exist $l \in \mathbb{Z}^+$ such that $\frac{l-2}{l-1} < \beta < \frac{l-1}{l}$. Define another activation function*

$$\hat{\sigma}_l(x) := \left(\sum_{j=0}^{l-1} \frac{1}{j!}h_j H_j(x/b)\right) + h_l^* z \quad with \quad z \sim \mathcal{N}(0, 1), \tag{16}$$

*where $h_j := \mathbb{E}_{z \sim \mathcal{N}(0,1)}[H_j(z)\sigma(bz)]$, and $h_l^* := \sqrt{\mathbb{E}_{z \sim \mathcal{N}(0,1)}[\sigma(bz)^2] - \sum_{j=0}^{l-1} h_j^2/(j!)}$ with $b := \sqrt{n/Tr(\boldsymbol{\Sigma})}$. Consider replacing the activation $\sigma(x)$ with the polynomial activation $\hat{\sigma}_l(x)$ after the training of the first layer $\hat{\boldsymbol{F}}$ as in (3). Then,*

  *(i) the training error $\mathcal{T}$ with activation $\sigma$, and that with the polynomial activation $\hat{\sigma}_l$, both converge in probability to the same value,*

  *(ii) the corresponding generalization errors $\mathcal{G}$ also converge in probability to the same value if an additional assumption (A.9) provided in Appendix B hold.*

*Proof.* Appendix C. □

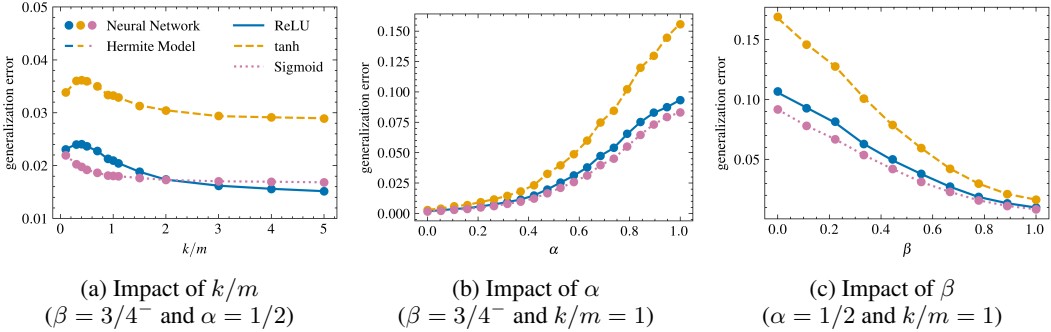

(a) Impact of $k/m$
($\beta = 3/4^-$ and $\alpha = 1/2$)

(b) Impact of $\alpha$
($\beta = 3/4^-$ and $k/m = 1$)

(c) Impact of $\beta$
($\alpha = 1/2$ and $k/m = 1$)

Figure 1: Generalization error comparison between neural network and the Hermite model. We set both the input dimension and the number of samples to $n = m = 1000$, with two Gaussian components ($\mathcal{C} = 2$) and covariance matrix ranks of $d_1 = d_2 = 1$. The mixture ratio for both components is set to $\rho_1 = \rho_2 = 0.5$, and a regularization constant of $\lambda = 1e - 4$ is applied. For the labels, we utilize $y = \text{ReLU}(\boldsymbol{\xi}^T \boldsymbol{x})$, and we limit the maximum degree of the Hermite polynomial to $l = 5$ for numerical stability. The figure presents averages from 20 Monte Carlo simulations.

Theorem 4 is pivotal as it establishes that, under specific conditions, the activation function $\sigma(x)$ in the neural network can be effectively substituted with a polynomial activation $\hat{\sigma}_l(x)$, which is constructed from Hermite polynomials (O'Donnell, 2014, Chapter 11.2) up to degree $l - 1$, without compromising training and generalization errors. The strength parameter $\beta = \frac{\log(\eta \|\boldsymbol{\Sigma}\|)}{\log(n)}$ plays a critical role in determining the necessary degree of the polynomial activation. Notably, Theorem 4 extends the equivalence results presented by Moniri et al. (2024) to encompass more general data scenarios. However, similar to the limitations identified in Moniri et al. (2024) regarding the maximal scale for the learning rate, Theorem 4 does not address the maximal value for the strength parameter ($\beta = 1$), in contrast to Theorem 3. While $\beta \to 1$ implies $l \to \infty$, we observe that a finite $l$ value is enough to achieve the equivalence of generalization errors in our numerical simulations for $\beta \approx 1$. Furthermore, the choice of Hermite polynomials is particularly useful due to their orthogonality properties (Lemma 9 in Appendix B) when applied to Gaussian inputs. Consequently, we refer to the resulting model as the equivalent "Hermite model":

$$\frac{1}{\sqrt{k}} \boldsymbol{w}^T \hat{\sigma}_l(\hat{\boldsymbol{F}} \boldsymbol{x}), \tag{17}$$

where $\hat{\sigma}_l$ is defined as a finite sum of Hermite polynomials scaled by coefficients $h_j$, supplemented by a Gaussian noise term that accounts for residuals. This equivalence significantly simplifies our analysis by transforming the nonlinear activation into a polynomial form, thereby enhancing the tractability of the model while maintaining its performance characteristics. The polynomial representation offers two key advantages: first, it has superior theoretical properties, such as easier performance characterization compared to general nonlinear forms; second, it defines an equivalence class of activation functions based on their polynomial coefficients. Activation functions with the same coefficients will yield identical performance outcomes, opening intriguing possibilities for future research. This equivalence class could be crucial in the search for optimal activation functions, enabling more targeted exploration of their effects on model performance. Overall, by leveraging this framework, we can more readily explore the implications of feature learning and structured data on neural network behavior.

## 6 SIMULATION RESULTS AND DISCUSSION

In this section, we present our numerical results and provide a detailed discussion of their implications. Each result illustrates the generalization errors corresponding to the parameter of interest for each specific plot. To showcase the applicability of our theoretical findings, we evaluate three widely used activation functions simultaneously: ReLU (rectified linear unit), tanh (hyperbolic tangent), and Sigmoid (logistic function). This comparison allows us to assess how different nonlinearities impact generalization behavior, validating our theoretical predictions and highlighting the practical relevance of our results in guiding activation function selection in neural network architectures.

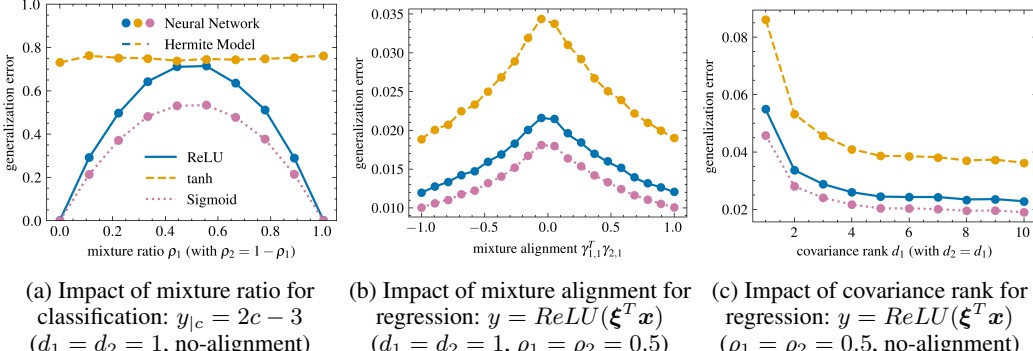

(a) Impact of mixture ratio for classification: $y_{|c} = 2c - 3$ ($d_1 = d_2 = 1$, no-alignment)

(b) Impact of mixture alignment for regression: $y = ReLU(\boldsymbol{\xi}^T \boldsymbol{x})$ ($d_1 = d_2 = 1$, $\rho_1 = \rho_2 = 0.5$)

(c) Impact of covariance rank for regression: $y = ReLU(\boldsymbol{\xi}^T \boldsymbol{x})$ ($\rho_1 = \rho_2 = 0.5$, no-alignment)

Figure 2: Impacts of properties of the Gaussian mixture data model on generalization performance. Here, we set the number of Gaussian components to $\mathcal{C} = 2$, with equal input dimensions and sample sizes of $m = n = k = 1000$. The parameters are configured with $\beta = 3/4^-$, $\alpha = 1/2$, $l = 4$, and a regularization constant of $\lambda = 1e - 4$. For (a) and (b), the eigenvalues of the covariance matrix (9) for each Gaussian component are fixed at $\theta_{1,1} = \theta_{2,1} = n^\beta$, while in (c), the eigenvalues $\{\theta_{c,i}\}_{i=1}^{d_c}$ are sampled uniformly from the interval $(0, n^\beta)$. The results displayed are averages from 20 Monte Carlo simulations, with data resampled for each run.

**Effect of model complexity —** Figure 1a demonstrates that the generalization errors of both the neural network and the equivalent Hermite model closely align across all values of $k/m$, reinforcing our theoretical findings. Supporting this, Figure 6 (given in Appendix D.3) reveals that the training errors for both models also match closely. However, since generalization performance is of greater interest than training performance, we will concentrate on generalization errors in the subsequent plots. It is worth noting that our remaining simulation results are presented for the case of $k/m = 1$, although similar outcomes can be observed for other $k/m$ ratios, indicating the robustness of our findings across different settings.

**High data spread instead of high learning rate for better performance —** In Figure 1b, we investigate the impact of $\alpha$ on generalization error while keeping $\beta = 3/4^-$ constant. Here, $\alpha$ influences the ratio of the learning rate $\eta$ to the data spread, i.e., the norm of the input covariance matrix $\|\boldsymbol{\Sigma}\|$, characterized by $\|\boldsymbol{\Sigma}\| \asymp n^{\beta(1-\alpha)}$ and $\eta \asymp n^{\beta\alpha}$. The results reveal that as $\alpha$ increases, the generalization error also rises, indicating that the strength of the data's structure is more beneficial for generalization performance than the strength of the learning rate in the first layer. This observation underscores the importance of structured data in enhancing model performance.

**Larger strength parameter $\beta$ for improved generalization —** In Figure 1c, we examine the effect of $\beta$ on generalization error while keeping $\alpha = 1/2$ constant. The parameter $\beta$ governs the product of the learning rate $\eta$ and the norm of the input covariance matrix $\|\boldsymbol{\Sigma}\|$ (a.k.a. data spread) through the relationship $\eta\|\boldsymbol{\Sigma}\| \asymp n^\beta$. The results indicate that as $\beta$ increases, generalization errors decrease, reflecting the benefits of more complex data distributions (larger $\|\boldsymbol{\Sigma}\|$) and higher learning rates associated with larger $\beta$ values. Note that while higher $\beta$ leads to better generalization in our setting, $\alpha$ value shapes the curve of the generalization error with respect to $\beta$, as illustrated in Appendix D.2.

**Properties of Gaussian mixture —** Next, we investigate how the properties of the Gaussian mixture data model (2) influence generalization performance. Figure 2 illustrates the generalization errors associated with various data characteristics. First, we explore the effects of imbalanced data through different mixture ratios in Figure 2a. While both ReLU and Sigmoid activations achieve zero error for completely imbalanced cases, the mixture ratio has minimal impact on generalization error with the tanh activation function due to its symmetry and the zero-mean input, which prevents bias toward any class. Next, we examine the significance of mixture alignment—specifically, the alignment between the spiked directions of two Gaussian components—on generalization performance in regression. As shown in Figure 2b, increasing alignment $|\boldsymbol{\gamma}_{1,1}^T \boldsymbol{\gamma}_{2,1}|$ towards 1 decreases generalization error since the Gaussian mixture reduces to a single Gaussian distribution when $|\boldsymbol{\gamma}_{1,1}^T \boldsymbol{\gamma}_{2,1}| = 1$.

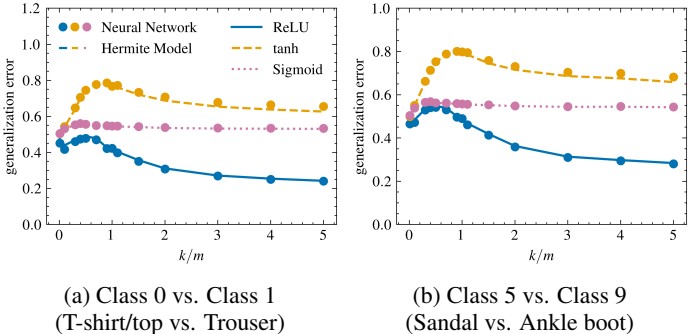

(a) Class 0 vs. Class 1
(T-shirt/top vs. Trouser)

(b) Class 5 vs. Class 9
(Sandal vs. Ankle boot)

Figure 3: Simulation results on Fashion-MNIST binary classification for $\|\mathbf{\Sigma}\| = n$ and $\eta = 1$. The data is generated from a conditional GAN trained on Fashion-MNIST dataset and pre-processed. For the pre-processing, the inputs from each class are demeaned, re-scaled and added noise such that assumptions (A.2)-(A.4) are satisfied. $m = 500$, $\lambda = 1e-4$ and $l = 5$. Details for the simulations and examples of input images after the pre-processing are provided in Appendix E.

This suggests that mixture data presents greater challenges than single Gaussian data. Finally, Figure 2c reveals that a higher effective rank ($d_c$) of the covariance matrix leads to decreased generalization error. This occurs because a higher rank, on average, leads to a larger maximum eigenvalue due to our random sampling of eigenvalues for the covariance matrices.

Collectively, the results in Figures 1 and 2 confirm that the generalization performances of neural networks align closely with those of the equivalent Hermite model, underscoring the strength of our theoretical findings across diverse scenarios.

**Towards theoretical results on real data —** In this work, we provide theoretical insights on realistic data, focusing on Gaussian mixture models, since previous studies (Seddik et al., 2020; Dandi et al., 2023b) have shown that data generated by generative adversarial networks (GANs) resemble Gaussian mixtures. We present simulation results from a conditional GAN (cGAN) (Mirza & Osindero, 2014) trained on the Fashion-MNIST dataset (Xiao et al., 2017), enabling us to generate samples conditioned on specific classes. This allows us to create two binary classification tasks: Class 0 (T-shirt/top) vs. Class 1 (Trouser) and Class 5 (Sandal) vs. Class 9 (Ankle boot). For this setting, our simulation results in Figure 3, reveal that a finite-degree Hermite model ($l = 5$) achieves nearly the same generalization error as the neural network. Therefore, our method allows for a direct examination of the neural network's activation function $\sigma$ via its equivalent Hermite model's activation function $\hat{\sigma}_l$. Additionally, we discuss the effects of learning rate $\eta$ in Appendix D.4, suggesting potential improvements in generalization error when $|\mathbf{\Sigma}| \asymp n$ and $\eta \asymp n$. Thus, extending our work to cover $\beta \in (1, 2]$ would be an intriguing direction for future research.

## 7 Conclusion

In this work, we have explored the behavior of two-layer neural networks after a single gradient step within the framework of the asymptotic proportional limit, specifically under the assumption of Gaussian mixture data. Our analysis provides a comprehensive understanding of how structured data and feature learning jointly influence the generalization performance of these networks. We have established that a conditional Gaussian model is equivalent to the two-layer neural network in terms of both training and generalization performance. Furthermore, we demonstrated that a finite-degree polynomial model can effectively approximate this conditional Gaussian model, thereby showing that polynomial models can also perform equivalently to neural networks. We also highlighted potential avenues for future research, particularly the extension of the range of $\beta$, which governs the joint scaling of feature learning and data spread. Our simulation results illustrate the impact of various properties within the Gaussian mixture data setting, reinforcing our theoretical findings. Importantly, we applied our theoretical results to a practical classification problem using the Fashion-MNIST dataset, where input images were generated through a cGAN. This work not only enhances our understanding of neural networks in structured data contexts but also sets the stage for further investigations into their performance on realistic datasets.

ACKNOWLEDGMENTS

We acknowledge that this work is supported in part by the TÜBİTAK-ARDEB 1001 program under project 124E063. We extend our gratitude to TÜBİTAK for its support. S.D. is supported by an AI Fellowship provided by Koç University & İş Bank Artificial Intelligence (KUIS AI) Research Center and a PhD Scholarship (BİDEB 2211) from TÜBİTAK.

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

## NOTE TO READERS OF THE PROOFS: HIGH-PROBABILITY EVENTS

In the following sections, we leverage high-probability bounds to establish our results, focusing on the core components of the proofs while omitting detailed tail bounds for brevity. We provide appropriate references for readers seeking comprehensive derivations. We are confident that the proofs, as presented, are both rigorous and accessible, adhering to the standards of a conference paper format. Furthermore, we plan to release an extended version of this work for future journal publication, which will include more detailed proofs and additional insights, thereby enriching the understanding of our findings and their implications in the field.

## A SPIKE+BULK DECOMPOSITION OF THE GRADIENT

Following Ba et al. (2022), we study the gradient $\boldsymbol{G}$ as follows:

$$\boldsymbol{G} := \frac{1}{m}\left(\frac{1}{\sqrt{k}}\left(\boldsymbol{w}\tilde{\boldsymbol{y}}^T - \frac{1}{\sqrt{k}}\boldsymbol{w}\boldsymbol{w}^T\sigma(\boldsymbol{F}\tilde{\boldsymbol{X}}^T)\right) \odot \sigma'(\boldsymbol{F}\tilde{\boldsymbol{X}}^T)\right)\tilde{\boldsymbol{X}}, \tag{18}$$

$$= \underbrace{\frac{\tilde{h}_1}{m\sqrt{k}}\boldsymbol{w}\tilde{\boldsymbol{y}}^T\tilde{\boldsymbol{X}}}_{\boldsymbol{A}} + \underbrace{\frac{1}{m\sqrt{k}}\left(\boldsymbol{w}\tilde{\boldsymbol{y}}^T \odot \sigma'_{\perp}(\boldsymbol{F}\tilde{\boldsymbol{X}}^T)\right)\tilde{\boldsymbol{X}}}_{\boldsymbol{B}} - \underbrace{\frac{1}{mk}\left(\boldsymbol{w}\boldsymbol{w}^T\sigma(\boldsymbol{F}\tilde{\boldsymbol{X}}^T) \odot \sigma'(\boldsymbol{F}\tilde{\boldsymbol{X}}^T)\right)\tilde{\boldsymbol{X}}}_{\boldsymbol{C}}, \tag{19}$$

where we use the orthogonal decomposition: $\sigma'(z) = \tilde{h}_1 + \sigma'_{\perp}(z)$ and $\tilde{h}_1 := \mathbb{E}_{z \sim \mathcal{N}(0,1)}[\sigma'(z)]$. Here, the expectation is over $z \sim \mathcal{N}(0,1)$ since for all $i \in \{1,\dots,k\}$, $\boldsymbol{f}_i^T\boldsymbol{x}$ has Gaussian distribution (conditioned on component $c$) for fixed $\boldsymbol{f}_i$ and we have $\mathbb{E}_{\boldsymbol{f}_i,\boldsymbol{x}}[\boldsymbol{f}_i^T\boldsymbol{x}] = 0$ and $\mathbb{E}_{\boldsymbol{f}_i,\boldsymbol{x}}[(\boldsymbol{f}_i^T\boldsymbol{x})^2] = 1$ by (A.3) and (A.6). Furthermore, $\tilde{h}_1 = \mathcal{O}(1)$ since $\sigma'(\cdot)$ has bounded derivatives by (A.8).

Then, we have $\boldsymbol{A} = \boldsymbol{u}\boldsymbol{v}^T$ for $\boldsymbol{u} := \tilde{h}_1\boldsymbol{w}$ and $\boldsymbol{v} := \tilde{\boldsymbol{X}}^T\tilde{\boldsymbol{y}}/(m\sqrt{k})$ while $\boldsymbol{\Delta} := \boldsymbol{B} - \boldsymbol{C}$, which gives us $\boldsymbol{G} = \boldsymbol{u}\boldsymbol{v}^T + \boldsymbol{\Delta}$. Here, $\boldsymbol{A}$ matrix represents the spike structure of the gradient while $\boldsymbol{\Delta}$ is called the bulk. Next, we show $\|\boldsymbol{\Delta}\| = \tilde{\mathcal{O}}(k^{-t})$ with high probability by studying the norms of $\boldsymbol{B}$ and $\boldsymbol{C}$, where $t := 1 - \beta(1-\alpha) \geq 0$. We start with bounding $\|\boldsymbol{B}\|$

$$\|\boldsymbol{B}\| \leq \frac{1}{m\sqrt{k}}\|\boldsymbol{w}\tilde{\boldsymbol{y}}^T \odot \sigma'_{\perp}(\boldsymbol{F}\tilde{\boldsymbol{X}}^T)\|\|\tilde{\boldsymbol{X}}\|, \tag{20}$$

$$= \frac{1}{m\sqrt{k}}\|\text{diag}(\boldsymbol{w})\sigma'_{\perp}(\boldsymbol{F}\tilde{\boldsymbol{X}}^T)\text{diag}(\tilde{\boldsymbol{y}})\|\|\tilde{\boldsymbol{X}}\|, \tag{21}$$

$$\leq \frac{1}{m\sqrt{k}}\|\boldsymbol{w}\|_{\infty}\|\tilde{\boldsymbol{y}}\|_{\infty}\|\sigma'_{\perp}(\boldsymbol{F}\tilde{\boldsymbol{X}}^T)\|\|\tilde{\boldsymbol{X}}\|, \tag{22}$$

for which we control the norms on the last line one-by-one as follows. First, we get

$$\|\boldsymbol{w}\|_{\infty} = \tilde{\mathcal{O}}(k^{-1/2}), \tag{23}$$

with high probability, due to sub-Gaussian tail bound (Vershynin, 2018, Proposition 2.5.2) and (A.6). Next, we have

$$\|\tilde{\boldsymbol{y}}\|_{\infty} = \tilde{\mathcal{O}}(1), \tag{24}$$

with high probability, due to Gaussian concentration of Lipschitz functions (Vershynin, 2018, Theorem 5.2.2) and since we can equivalently write $\tilde{y}_i = \sigma_*(z_i)$ for $z_i \sim \mathcal{N}(0, C_i)$ with $C_i = \tilde{\mathcal{O}}(1)$ by (A.3)-(A.5) while $\sigma_*$ is a Lipschitz function by (A.7). Next, we get bounds on $\|\boldsymbol{F}\|$ and $\|\tilde{\boldsymbol{X}}\|$ as

$$\|\boldsymbol{F}\| = \tilde{\mathcal{O}}\left(1\right), \quad \text{and} \quad \|\tilde{\boldsymbol{X}}\|/\|\boldsymbol{\Sigma}^{1/2}\| = \tilde{\mathcal{O}}\left(k^{1/2}\right), \tag{25}$$

with high probability, due to concentration of norms of sub-Gaussian matrices (Vershynin, 2018, Theorem 4.4.5). The bound for $\|\boldsymbol{F}\|$ is due to (A.6) and $\text{Tr}(\boldsymbol{\Sigma}) \asymp k$ by (A.4). The effect of the mixture (2) is handled by considering $\tilde{\boldsymbol{X}}$ as a concatenation of Gaussian matrices. Note that $k, n, m$ can used interchangeably in the bounds due to (A.1). Using (25), we get

$$\|\sigma'_{\perp}(\boldsymbol{F}\tilde{\boldsymbol{X}}^T)\| = \tilde{\mathcal{O}}\left(k^{1/2+(1-t)/2}\right), \tag{26}$$

with high probability. Here, the rows of $\sigma'_\perp(F\tilde{X}^T)^T$ are independent sub-Gaussian vectors, which follows from Gaussian concentration of Lipschitz functions (Vershynin, 2018, Theorem 5.2.2) and the boundedness of derivatives of $\sigma'_\perp$ by (A.8). Thus, we get $\|\sigma'_\perp(F\tilde{X}^T)\| \leq \|F\|\|\tilde{X}\|$ polylog $k$ with high probability using the concentration of the norm of a matrix with independent sub-Gaussian rows (Vershynin, 2010, Theorem 5.39 and Equation 5.26). Furthermore, $\|\Sigma\| = \tilde{\mathcal{O}}(k^{1-t})$ by (A.2), which makes $\|\tilde{X}\| = \tilde{\mathcal{O}}\left(k^{1/2+(1-t)/2}\right)$ with high probability due to (25) and allow us to reach (26).

Putting everything together, we get $\|B\| = \tilde{\mathcal{O}}\left(k^{-t}\right)$ with high probability.

Similarly, we focus on $\|C\|$,

$$\|C\| \leq \frac{1}{mk}\left\|ww^T\sigma(F\tilde{X}^T) \odot \sigma'(F\tilde{X}^T)\right\| \|\tilde{X}\|, \tag{27}$$

$$= \frac{1}{mk}\left\|\operatorname{diag}(w)\sigma'(F\tilde{X}^T)\operatorname{diag}(w^T\sigma(F\tilde{X}^T))\right\| \|\tilde{X}\|, \tag{28}$$

$$\leq \frac{1}{mk}\|w\|_\infty\|w^T\sigma(F\tilde{X}^T)\|_\infty\|\sigma'(F\tilde{X}^T)\|\|\tilde{X}\|, \tag{29}$$

for which we find high probability bounds for the norms on the last line one-by-one as follows. Note that bounds for $\|w\|_\infty$ and $\|\tilde{X}\|$ are found before in (23) and (25), respectively. Thus, we study the remaining two terms starting with $\|\sigma'(F\tilde{X}^T)\|$ as follows:

$$\|\sigma'(F\tilde{X}^T)\| = \tilde{\mathcal{O}}(k), \tag{30}$$

with high probability, which follows from $\|\sigma'(F\tilde{X}^T)\| \leq \|\sigma'_\perp(F\tilde{X}^T)\| + \|\tilde{h}_1\mathbf{1}_{k\times m}\|$, where $\mathbf{1}_{k\times m}$ denotes all ones matrix of dimensions $k \times m$. Here, $\tilde{h}_1 = \mathcal{O}(1)$ as discussed at the beginning of the section and a bound for $\|\sigma'_\perp(F\tilde{X}^T)\|$ is given in (26). For the last term, we have

$$\|w^T\sigma(F\tilde{X}^T)\|_\infty = \tilde{\mathcal{O}}\left(k^{(1-t)/2}\right), \tag{31}$$

with high probability, due to the Gaussian concentration of Lipschitz functions (Vershynin, 2018, Theorem 5.2.2). Note that $\|w\| = \tilde{\mathcal{O}}(1)$ with high probability due to the concentration of the norm of a Gaussian vector (Vershynin, 2018, Theorem 3.1.1) and (A.6). Combining these bounds, we get $\|C\| = \tilde{\mathcal{O}}\left(k^{-t}\right)$ with high probability.

Using the bounds on $\|B\|$ and $\|C\|$, we reach

$$\|\Delta\| = \|B - C\| \leq \|B\| + \|C\| = \tilde{\mathcal{O}}\left(k^{-t}\right), \tag{32}$$

with high probability.

Similarly, we also get

$$\|u\| = |\tilde{h}_1|\|w\| = \tilde{\mathcal{O}}(1), \quad \text{and} \quad \|v\| \leq \frac{1}{m\sqrt{k}}\|\tilde{X}\|\|\tilde{y}\| = \tilde{\mathcal{O}}\left(k^{-t/2}\right), \tag{33}$$

with high probability using the found norm bounds, which completes our proof.

## B    PROOF OF CONDITIONAL GAUSSIAN EQUIVALENCE (THEOREM 3)

Here, we prove the conditional Gaussian equivalence under Gaussian mixtures data setting following the proof technique in Dandi et al. (2023a). We first provide the following lemma describing a conditional central limit theorem (CLT) in our setting.

**Lemma 5** (Conditional CLT). *For any Lipschitz function $\psi : \mathbb{R}^2 \to \mathbb{R}$, $\forall c \in \{1, \ldots, \mathcal{C}\}$ and $\forall \kappa_c \in \mathbb{R}^{d_c+1}$,*

$$\lim_{n,k\to\infty} \sup_{\tilde{w}\in\mathcal{S}_k,\|\xi\|=1/\|\Sigma^{1/2}\|} \left|\mathbb{E}\left[\psi\left(\tilde{w}^T\phi(x),\xi^Tx\right) \mid c,\kappa_c\right] - \mathbb{E}\left[\psi\left(\tilde{w}^T\hat{\phi}(x),\xi^Tx\right) \mid c,\kappa_c\right]\right| = 0, \tag{34}$$

*where $\mathcal{S}_k := \{\tilde{w} \in \mathbb{R}^k \mid \|\tilde{w}\| \leq C_1, \quad \|\tilde{w}\|_\infty = C_2/k^{-\epsilon}\}$ for some $C_1, C_2, \epsilon > 0$.*

*Proof.* Due to the decomposition of $\hat{\boldsymbol{F}}\boldsymbol{x}$ (Lemma 2), we can equivalently study $\sigma(\boldsymbol{a}_{|\boldsymbol{\kappa}_c} + \boldsymbol{F}^{\perp}\boldsymbol{z}^{\perp})$ instead of $\sigma(\hat{\boldsymbol{F}}\boldsymbol{x}_{|c})$. To proceed, we define $\tilde{\boldsymbol{F}} := \boldsymbol{F}^{\perp}(\boldsymbol{I}_n - \boldsymbol{\Gamma}_c(\boldsymbol{\Gamma}_c^T\boldsymbol{\Gamma}_c)^{-1}\boldsymbol{\Gamma}_c^T)$ in order to focus on the following equivalent features: $\sigma(\boldsymbol{a}_{|\boldsymbol{\kappa}_c} + \tilde{\boldsymbol{F}}\tilde{\boldsymbol{z}})$ for $\tilde{\boldsymbol{z}} \sim \mathcal{N}(0, \boldsymbol{I}_n)$. Then, we can define the following neuron-wise activation functions

$$\sigma_{i|\kappa_c}(u) := \sigma\left(a_{i|\kappa_c} + b_i u\right) - \mathbb{E}_{\hat{u}\sim\mathcal{N}(0,1)}\left[\sigma\left(a_{i|\kappa_c} + b_i\hat{u}\right)\right], \tag{35}$$

where $b_i := \|\tilde{\boldsymbol{f}}_i\| > 0$ for all $i \in \{1, \ldots, k\}$. Now, we define $\tilde{\tilde{\boldsymbol{F}}} := [\tilde{\tilde{\boldsymbol{f}}}_1, \ldots, \tilde{\tilde{\boldsymbol{f}}}_k]^T$ where $\tilde{\tilde{\boldsymbol{f}}}_i := \tilde{\boldsymbol{f}}_i/b_i$. Thus, we focus on the feature mapping of the form $\sigma_{i|\kappa_c}(\tilde{\tilde{\boldsymbol{f}}}_i^T\tilde{\boldsymbol{z}})$, which is equivalent to random features mapping (Rahimi & Recht, 2007; Hu & Lu, 2023) with activation functions differing across neurons. The one-dimensional CLT for random features (Goldt et al., 2022; Hu & Lu, 2023; Montanari & Saeed, 2022) holds even when the activation functions differ across neurons as observed by Dandi et al. (2023b;a). Therefore, we check the assumptions used in showing the one-dimensional CLT for random features in Hu & Lu (2023). Here, the following events hold with high probability:

- $\sup_{i,j\in\{1,\ldots,k\}} \left|\tilde{\tilde{\boldsymbol{f}}}_i^T\tilde{\tilde{\boldsymbol{f}}}_j - \delta_{ij}\right| = \tilde{\mathcal{O}}(1/k^{1/2})$,

- $\|\tilde{\tilde{\boldsymbol{F}}}\| = \tilde{\mathcal{O}}(1)$,

which follow from the bounds for $\|\boldsymbol{F}\|$ and $\|\boldsymbol{\Delta}\|$ provided in Appendix A and the fact that $\|\boldsymbol{I}_n - \boldsymbol{\Gamma}_c(\boldsymbol{\Gamma}_c^T\boldsymbol{\Gamma}_c)^{-1}\boldsymbol{\Gamma}_c^T\| \leq 1 + \|\boldsymbol{\Gamma}_c(\boldsymbol{\Gamma}_c^T\boldsymbol{\Gamma}_c)^{-1}\boldsymbol{\Gamma}_c^T\| = \mathcal{O}(1)$ by definition. Furthermore, $\mathbb{E}_{u\sim\mathcal{N}(0,1)}\sigma_{i|\kappa_c}(u) = 0$. Therefore, we can utilize the Lemma 2 in Hu & Lu (2023). Note that the odd activation function assumption, which is used to match covariances in Hu & Lu (2023), can be dropped here since $\phi(\boldsymbol{x})$ and $\hat{\phi}(\boldsymbol{x})$ have exactly the same mean and covariance. Finally, to cover the effect of the second parameter of the test function $\psi$, we set $\tilde{\tilde{\boldsymbol{f}}}_0 := \boldsymbol{\Sigma}_c^{1/2}\boldsymbol{\xi}/\left\|\boldsymbol{\Sigma}_c^{1/2}\boldsymbol{\xi}\right\|$ and use Theorem 2 in Hu & Lu (2023). $\qquad\square$

Lemma 5 is useful for showing performance-wise equivalence of the two feature maps since it states that the conditional expectation of any test function for the original feature map $\phi(\boldsymbol{x})$ is equal to that for the equivalent conditional feature map $\hat{\phi}(\boldsymbol{x}; c, \boldsymbol{\kappa}_c)$ in the limit. To apply Lemma 5 in our results about training and generalization error, we need to assume $\hat{\boldsymbol{w}}/\sqrt{k}$ in (5) to satisfy $\hat{\boldsymbol{w}}/\sqrt{k} \in \mathcal{S}_k$ with high probability, where $\mathcal{S}_k$ is defined in the lemma. Note that $\mathcal{S}_k$ is used in the proof of Lemma 5 when we utilize the CLT results from Hu & Lu (2023). One can show that $\hat{\boldsymbol{w}}/\sqrt{k} \in \mathcal{S}_k$ holds with high probability similar to Lemma 17-18 and Lemma 23 in Hu & Lu (2023), which is omitted here. Alternatively, one can include $\hat{\boldsymbol{w}}/\sqrt{k} \in \mathcal{S}_k$ as a constraint into the optimization objective of $\hat{\boldsymbol{w}}$ in (5). Either way, we continue assuming this condition holds.

Recall the original feature map and the equivalent conditional feature map at this point:

$$\phi(\boldsymbol{x}) := \sigma(\hat{\boldsymbol{F}}\boldsymbol{x}), \tag{36}$$

$$\hat{\phi}(\boldsymbol{x}; c, \boldsymbol{\kappa}_c) := \boldsymbol{\nu}(c, \boldsymbol{\kappa}_c) + \boldsymbol{\Psi}(c, \boldsymbol{\kappa}_c)\boldsymbol{z}^{\perp} + \boldsymbol{\Phi}(c, \boldsymbol{\kappa}_c)\boldsymbol{g}, \tag{37}$$

where $\boldsymbol{z} = \boldsymbol{\Sigma}_c^{-1/2}\boldsymbol{x}_{|c} = \boldsymbol{\Gamma}_c\boldsymbol{\kappa}_c + \boldsymbol{z}^{\perp}$, and

$$\boldsymbol{\nu}(c, \boldsymbol{\kappa}_c) := \mathbb{E}\left[\sigma(\hat{\boldsymbol{F}}\boldsymbol{x}) \mid c, \boldsymbol{\kappa}_c\right], \qquad \boldsymbol{\Psi}(c, \boldsymbol{\kappa}_c) := \mathbb{E}\left[\sigma(\hat{\boldsymbol{F}}\boldsymbol{x})(\boldsymbol{z}^{\perp})^T \mid c, \boldsymbol{\kappa}_c\right], \tag{38}$$

$$\boldsymbol{\Phi}(c, \boldsymbol{\kappa}_c) := \mathrm{Cov}\left(\sigma(\hat{\boldsymbol{F}}\boldsymbol{x}) \mid c, \boldsymbol{\kappa}_c\right) - \boldsymbol{\Psi}(c, \boldsymbol{\kappa}_c)\boldsymbol{\Psi}(c, \boldsymbol{\kappa}_c)^T, \qquad \boldsymbol{g} \sim \mathcal{N}(0, \boldsymbol{I}_n). \tag{39}$$

In the rest of the proof, we use the universality result for Gaussian mixtures in generalized linear models by Dandi et al. (2023b), which is an extension of the universality result by Montanari & Saeed (2022) to mixture settings. There are a couple of points to be modified in order to apply their results in our case. First, while Dandi et al. (2023b) assumes a conditional CLT (conditioned on component index $c$) in their Assumption 4, we replace it with Lemma 5 which is another conditional CLT (conditioned on $c$ and $\boldsymbol{\kappa}_c$ together). Furthermore, Dandi et al. (2023b) supposes bounded mean in their Assumption 2, which translates to $\|\boldsymbol{\nu}(c, \boldsymbol{\kappa}_c)\| \leq C$ for some $C > 0$. In our case, this

assumption does not hold, which is the main challenge. To overcome this challenge, we study the conditional mean of the features $\boldsymbol{\nu}(c, \boldsymbol{\kappa}_c)$ and the demeaned features $\phi(\boldsymbol{x}) - \boldsymbol{\nu}(c, \boldsymbol{\kappa}_c)$ separately. About the demeaned features $\phi(\boldsymbol{x}) - \boldsymbol{\nu}(c, \boldsymbol{\kappa}_c)$, we have the following lemma similar to Lemma 20-21 in Dandi et al. (2023a).

**Lemma 6.** *Under our assumptions, for a given $\boldsymbol{\kappa}_c, c$,*

(i) *the random vector $\phi(\boldsymbol{x}) - \boldsymbol{\nu}(c, \boldsymbol{\kappa}_c)$ is sub-Gaussian with sub-Gaussian norm independent of $\boldsymbol{\kappa}_c, c$ and dimensions $n, k$.*

(ii) *the matrix $\bar{\bar{\boldsymbol{\Phi}}}$, each row of which is a sample of $\phi(\boldsymbol{x}) - \boldsymbol{\nu}(c, \boldsymbol{\kappa}_c)$, satisfies $P\left(\left\|\bar{\bar{\boldsymbol{\Phi}}}\right\| \geq C_1 \sqrt{k}\right) \leq 2 \exp(-C_2 k)$ for some $C_1, C_2 > 0$.*

*Proof.* Here, (i) follows from the Gaussian concentration of Lipschitz functions (Vershynin, 2018, Theorem 5.2.2) and the boundedness of derivatives of $\sigma$.

(ii) is due to (i) and the concentration of the norms of matrices with independent sub-Gaussian rows (Vershynin, 2010, Theorem 5.39 and Equation 5.26). □

Next, we focus on $\boldsymbol{\nu}(c, \boldsymbol{\kappa}_c)$. To relax the assumption on its norm, we have the following lemma similar to Lemma 22 in Dandi et al. (2023a).

**Lemma 7.** *Suppose our assumptions. Let $\hat{\boldsymbol{w}}$ be as defined in (5). Define $\boldsymbol{\nu}(\boldsymbol{x}) := \boldsymbol{\nu}(c, \boldsymbol{\kappa}_c)$ where $c$ and $\boldsymbol{\kappa}_c$ are depending on $\boldsymbol{x}$ as defined in Lemma 2. Then, the following holds with high probability*

$$\frac{1}{m} \sum_{i=1}^{m} \left(\frac{1}{\sqrt{k}} \hat{\boldsymbol{w}}^T \boldsymbol{\nu}(\boldsymbol{x}_i)\right)^2 = \tilde{\mathcal{O}}(1). \tag{40}$$

*Proof.* First, recall that

$$\hat{\boldsymbol{w}} := \arg\min_{\boldsymbol{w}} \frac{1}{2m} \sum_{i=1}^{m} \left(y_i - \frac{1}{\sqrt{k}} \boldsymbol{w}^T \phi(\boldsymbol{x}_i)\right)^2 + \frac{\lambda}{2} \|\boldsymbol{w}\|^2, \tag{41}$$

Then, we have

$$\frac{1}{2m} \sum_{i=1}^{m} \left(y_i - \frac{1}{\sqrt{k}} \hat{\boldsymbol{w}}^T \phi(\boldsymbol{x}_i)\right)^2 + \frac{\lambda}{2} \|\hat{\boldsymbol{w}}\|^2 \leq \frac{1}{2m} \sum_{i=1}^{m} y_i^2 = \tilde{\mathcal{O}}(1), \tag{42}$$

where the first step is due to $\hat{\boldsymbol{w}}$ being the optimal solution of (41) while the second step is due to $|y_i| = \tilde{\mathcal{O}}(1)$ with high probability as mentioned in Appendix A. This leads to

$$\frac{1}{2m} \sum_{i=1}^{m} \left(\frac{1}{\sqrt{k}} \hat{\boldsymbol{w}}^T \phi(\boldsymbol{x}_i)\right)^2 \leq \frac{1}{m} \sum_{i=1}^{m} \left(\frac{1}{\sqrt{k}} \hat{\boldsymbol{w}}^T \phi(\boldsymbol{x}_i)\right) y_i, \tag{43}$$

$$\frac{1}{2m} \sum_{i=1}^{m} \left(\frac{1}{\sqrt{k}} \hat{\boldsymbol{w}}^T \phi(\boldsymbol{x}_i)\right)^2 \overset{(i)}{\leq} \sqrt{\frac{1}{m} \sum_{i=1}^{m} \left(\frac{1}{\sqrt{k}} \hat{\boldsymbol{w}}^T \phi(\boldsymbol{x}_i)\right)^2} \sqrt{\frac{1}{m} \sum_{i=1}^{m} y_i^2}, \tag{44}$$

$$\sqrt{\frac{1}{m} \sum_{i=1}^{m} \left(\frac{1}{\sqrt{k}} \hat{\boldsymbol{w}}^T \phi(\boldsymbol{x}_i)\right)^2} \overset{(ii)}{\leq} 2\sqrt{\frac{1}{m} \sum_{i=1}^{m} y_i^2}, \tag{45}$$

$$\frac{1}{m} \sum_{i=1}^{m} \left(\frac{1}{\sqrt{k}} \hat{\boldsymbol{w}}^T \phi(\boldsymbol{x}_i)\right)^2 \overset{(iii)}{=} \tilde{\mathcal{O}}(1), \tag{46}$$

$$\frac{1}{m} \sum_{i=1}^{m} \left(\frac{1}{\sqrt{k}} \hat{\boldsymbol{w}}^T \boldsymbol{\nu}(c, \boldsymbol{\kappa}_c)\right)^2 \overset{(iv)}{=} \tilde{\mathcal{O}}(1), \tag{47}$$

where (i) is due to the Cauchy–Schwarz inequality; (ii) follows from basic algebraic manipulation; (iii) is due to $|y_i| = \tilde{\mathcal{O}}(1)$ with high probability; finally, (iv) follows from Lemma 6. □

Now, we can utilize the universality of training error for Gaussian mixtures (Dandi et al., 2023b, Theorem 1 and Corollary 2) with the following modifications:

(i) Their Assumption 1 (Loss and regularization) holds as is in our setting.

(ii) Their Assumption 2 (Boundedness and concentration) requires boundedness for $\|\boldsymbol{\nu}(c, \boldsymbol{\kappa}_c)\|$. Thus, we need to relax it as mentioned before. The assumption is used for free energy approximation which directly follows (unchanged) from Montanari & Saeed (2022). The corresponding assumption in Montanari & Saeed (2022) is Assumption 5, which is used for their Lemma 5 and 6. Our Lemma 6 and 7 ensures that Lemma 5 and 6 in Montanari & Saeed (2022) hold for the case of unbounded and variable means $\boldsymbol{\nu}(c, \boldsymbol{\kappa}_c)$ across samples.

(iii) We have a modification to handle their Assumption 3 (Labels). The assumption does not directly allow labels $y$ to depend on $\boldsymbol{x}$. However, such a dependence can be incorporated by considering $[\phi(\boldsymbol{x}), \boldsymbol{x}] \in \mathbb{R}^{k+n}$ as the input of the generalized linear model in Dandi et al. (2023b) and constraining the last $n$ parameters of the model to be 0. Note that Lemma 5 (conditional CLT) includes $\boldsymbol{\xi}^T \boldsymbol{x}$ as the second parameter of the test function, which makes the CLT valid for such an input. Thus, Lemma 5 covers the dependence of labels to $\boldsymbol{x}$.

(iv) Their Assumption 4 (CLT) holds in our case due to our Lemma 5 (conditional CLT) and the law of total expectation.

For the universality of the generalization error, we utilize Theorem 4 in Dandi et al. (2023b), which requires the following additional assumption:

(A.9) Define a perturbed optimization objective

$$q_m(s) := \min_{\boldsymbol{w} \in \mathcal{S}_k} \frac{1}{2m} \sum_{i=1}^m \left( y_i - \frac{1}{\sqrt{k}} \boldsymbol{w}^T \sigma(\hat{\boldsymbol{F}} \boldsymbol{x}_i) \right)^2 + \frac{\lambda}{2} \|\boldsymbol{w}\|^2 + s\mathcal{G}(\boldsymbol{w}), \qquad (48)$$

where $s \in \mathbb{R}$ and $\mathcal{G}(\boldsymbol{w})$ is the generalization error defined in (7) when the second layer weight is $\boldsymbol{w}$. Then, there exist $s^* > 0$ such that, for $s \in [-s^*, s^*]$, the function $q_m(s)$ converges pointwise to a function $q(s)$ that is differentiable at $s = 0$.

Note that the additional assumption (A.9) corresponds to Assumption 5 in Dandi et al. (2023b), which is used to prove the equivalence of generalization errors using a convexity-based argument. Furthermore, assumptions similar to (A.9) can be found in Hu & Lu (2023) (see their Assumption 9) and Montanari & Saeed (2022) (see their Theorem 3). Finally, we observe that Assumption 6 in Dandi et al. (2023b) trivially holds in our case since we have a single minimizer $\hat{\boldsymbol{w}}$ in (5) and it can be simply found by ridge regression.

## C  PROOF OF THEOREM 4

In this section, we prove the equivalence of the two-layer neural network after training with one gradient step to the Hermite model. First of all, recall the bulk+structure decomposition of $\hat{\boldsymbol{F}}\boldsymbol{x}$ for the case of the input $\boldsymbol{x}_{|c}$ on $c$-th Gaussian,

$$\hat{\boldsymbol{F}}\boldsymbol{\Sigma}_c^{1/2}\boldsymbol{z} = \underbrace{(\boldsymbol{F} + \eta\boldsymbol{\Delta})\boldsymbol{z}^{\perp}}_{\boldsymbol{F}^{\perp}\boldsymbol{z}^{\perp}\ (\textbf{Bulk})} + \underbrace{\hat{\boldsymbol{F}}\boldsymbol{\Sigma}_c^{1/2}\boldsymbol{\Gamma}_c\boldsymbol{\kappa}_c}_{\boldsymbol{a}_{|\boldsymbol{\kappa}_c}\ (\textbf{Structure})}, \qquad (49)$$

where $\boldsymbol{z} = \boldsymbol{\Sigma}_c^{-1/2}\boldsymbol{x}_{|c} \sim \mathcal{N}(0, \boldsymbol{I}_n)$ and we use the orthogonal decomposition: $\boldsymbol{z} = \boldsymbol{\Gamma}_c\boldsymbol{\kappa}_c + \boldsymbol{z}^{\perp}$, with $\boldsymbol{\Gamma}_c := [\boldsymbol{v}, \boldsymbol{\gamma}_{c,1}, \boldsymbol{\gamma}_{c,2}, \ldots, \boldsymbol{\gamma}_{c,d_c}]$, $\boldsymbol{\kappa}_c := (\boldsymbol{\Gamma}_c^T\boldsymbol{\Gamma}_c)^{-1}\boldsymbol{\Gamma}_c^T\boldsymbol{z}$ and $\boldsymbol{z}^{\perp} := (\boldsymbol{I}_n - \boldsymbol{\Gamma}_c(\boldsymbol{\Gamma}_c^T\boldsymbol{\Gamma}_c)^{-1}\boldsymbol{\Gamma}_c^T)\boldsymbol{z}$.

Then, instead of $\sigma(\hat{\boldsymbol{F}}\boldsymbol{x}_{|c})$, we focus on the following equivalent features: $\sigma(\boldsymbol{a}_{|\boldsymbol{\kappa}_c} + \tilde{\boldsymbol{F}}\tilde{\boldsymbol{z}})$ where $\tilde{\boldsymbol{z}} \sim \mathcal{N}(0, \boldsymbol{I}_n)$ and $\tilde{\boldsymbol{F}} := \boldsymbol{F}^{\perp}(\boldsymbol{I}_n - \boldsymbol{\Gamma}_c(\boldsymbol{\Gamma}_c^T\boldsymbol{\Gamma}_c)^{-1}\boldsymbol{\Gamma}_c^T)$. This can be equivalently written as follows for each neuron $i$:

$$\sigma\left(a_{i|\kappa_c} + b_i u_i\right), \qquad (50)$$

where $b_i := \|\tilde{\boldsymbol{f}}_i\| > 0$ and $u_i := (\boldsymbol{f}_i^{\perp})^T \boldsymbol{z}^{\perp}/b_i \sim \mathcal{N}(0,1)$ for all $i \in \{1, \ldots, k\}$. Furthermore, $(u_i, u_j)$ is jointly Gaussian with $\mathbb{E}[u_i u_j] = \tilde{\boldsymbol{f}}_i^T \tilde{\boldsymbol{f}}_j/(b_i b_j)$ for all $i \neq j$.

For the rest of this proof, our task is to show that the polynomial activation $\hat{\sigma}_l(a_{i|\kappa_c} + b_i u_i)$, defined in (16), has conditional mean and covariance that are equivalent to those of $\sigma(a_{i|\kappa_c} + b_i u_i)$.

First, we start with the following lemma, allowing us to decompose $\sigma(a_{i|\kappa_c} + b_i u_i)$. Note that $\beta \le 1/2$ reduces the equivalent activation $\hat{\sigma}_l$ to the noisy linear activation (model) that was studied in prior work (Hu & Lu, 2023; Moniri et al., 2024), so we focus on the case of $\beta > 1/2$ in our proof, which corresponds to $l > 2$, i.e., beyond the linear case.

**Lemma 8.** *If $\frac{1}{2} \le \frac{l-2}{l-1} < \beta < \frac{l-1}{l}$, then $|a_{i|\kappa_c}| = \tilde{\mathcal{O}}\left(1/k^{1-\beta}\right)$ and $|a_{i|\kappa_c}|^l = \tilde{\mathcal{O}}(1/k^{1+\epsilon})$ hold with high probability, where $\epsilon := l(1-\beta) - 1 > 0$.*

*Proof.* Recall that $a_{i|\kappa_c} := \hat{\boldsymbol{f}}_i^T \boldsymbol{\Sigma}_c^{1/2} \boldsymbol{\Gamma}_c \boldsymbol{\kappa}_c$. Then, we have the following with high probability

$$|a_{i|\kappa_c}| \le \left| \left( \hat{\boldsymbol{f}}_i^T \left( \boldsymbol{\Sigma}_c^{1/2} - \boldsymbol{I}_n \right) + \eta u_i \boldsymbol{v}^T \right) \boldsymbol{\Gamma}_c \boldsymbol{\kappa}_c \right| + \left| (\boldsymbol{f}_i^\perp)^T \boldsymbol{\Gamma}_c \boldsymbol{\kappa}_c \right|, \tag{51}$$

$$\le \left( \left\| \hat{\boldsymbol{f}}_i^T \left( \boldsymbol{\Sigma}_c^{1/2} - \boldsymbol{I}_n \right) \right\| + \left\| \eta u_i \boldsymbol{v}^T \right\| \right) \|\boldsymbol{\Gamma}_c \boldsymbol{\kappa}_c\| + \left| (\boldsymbol{f}_i^\perp)^T \boldsymbol{\Gamma}_c \boldsymbol{\kappa}_c \right|, \tag{52}$$

$$\le \left( \left\| \left( \boldsymbol{f}_i^\perp + \eta u_i \boldsymbol{v} \right)^T \left( \boldsymbol{\Sigma}_c^{1/2} - \boldsymbol{I}_n \right) \right\| + \left\| \eta u_i \boldsymbol{v}^T \right\| \right) \|\boldsymbol{\Gamma}_c \boldsymbol{\kappa}_c\| + \left| (\boldsymbol{f}_i^\perp)^T \boldsymbol{\Gamma}_c \boldsymbol{\kappa}_c \right|, \tag{53}$$

$$\le \left( \left\| (\boldsymbol{f}_i^\perp)^T \left( \boldsymbol{\Sigma}_c^{1/2} - \boldsymbol{I}_n \right) \right\| + \eta \left\| u_i \boldsymbol{v}^T \left( \boldsymbol{\Sigma}_c^{1/2} - \boldsymbol{I}_n \right) \right\| + \eta \left\| u_i \boldsymbol{v}^T \right\| \right) \|\boldsymbol{\Gamma}_c \boldsymbol{\kappa}_c\| + \left| (\boldsymbol{f}_i^\perp)^T \boldsymbol{\Gamma}_c \boldsymbol{\kappa}_c \right|, \tag{54}$$

where we apply the triangle inequality and Cauchy–Schwarz inequality repeatedly to reach the last line. Then, we study each of the terms in the last line by using the following facts:

- $\boldsymbol{z} \sim \mathcal{N}(0, \boldsymbol{I}_n)$ by definition (49),

- $\|\boldsymbol{\Gamma}_c(\boldsymbol{\Gamma}_c^T \boldsymbol{\Gamma}_c)^{-1}\boldsymbol{\Gamma}_c^T\|_F = \tilde{\mathcal{O}}(1)$ by assumption (A.4) and its definition (49),

- $\boldsymbol{f}_i \sim \mathcal{N}(0, \boldsymbol{I}_n/\text{Tr}(\boldsymbol{\Sigma}))$ by assumption (A.6),

- $\text{Tr}(\boldsymbol{\Sigma}) = \mathcal{O}(k)$ and $\text{Tr}(\boldsymbol{\Sigma}_c - \boldsymbol{I}_n) = \tilde{\mathcal{O}}(k^{\beta(1-\alpha)})$ by assumptions (A.2) and (A.4),

- $k$ and $n$ can used interchangeably since $k/n \in \mathbb{R}$ by (A.1).

First, we observe that $\|\boldsymbol{\Gamma}_c \boldsymbol{\kappa}_c\|^2 = \boldsymbol{z}^T \boldsymbol{\Gamma}_c(\boldsymbol{\Gamma}_c^T \boldsymbol{\Gamma}_c)^{-1}\boldsymbol{\Gamma}_c^T \boldsymbol{z} = \tilde{\mathcal{O}}(1)$ holds with high probability, which follows from Hanson-Wright inequality (Vershynin, 2018, Theorem 6.2.1).

Then, we have $\eta\|u_i \boldsymbol{v}^T\| = \tilde{\mathcal{O}}(n^{\beta-1})$ and $\eta\|u_i \boldsymbol{v}^T(\boldsymbol{\Sigma}_c^{1/2} - \boldsymbol{I}_n)\| = \tilde{\mathcal{O}}(k^{\beta-1})$ holding with high probability as explained in the following. Here, $|u_i| = |\tilde{h}_1 w_i| = \tilde{\mathcal{O}}(k^{-1/2})$ with high probability since $w_i \sim \mathcal{N}(0, 1/k)$ by assumption (A.6). Furthermore, $\|\boldsymbol{v}\| = \tilde{\mathcal{O}}(k^{(\beta(1-\alpha)-1)/2})$ with high probability by Lemma 1 while $\eta = \mathcal{O}(k^{\beta\alpha})$ and $\|\boldsymbol{\Sigma}_c^{1/2} - \boldsymbol{I}_n\| = \mathcal{O}(k^{\beta(1-\alpha)/2})$ by (A.2).

Next, we get $\|(\boldsymbol{f}_i^\perp)^T(\boldsymbol{\Sigma}_c^{1/2} - \boldsymbol{I}_n)\| = \tilde{\mathcal{O}}(k^{\beta-1})$ with high probability. Here, first recall that $\boldsymbol{f}_i^\perp := \boldsymbol{f}_i + \eta\boldsymbol{\Delta}_i$, then we have $\eta\|\boldsymbol{\Delta}_i\| \le \eta\|\boldsymbol{\Delta}\|$ and $\eta\|\boldsymbol{\Delta}\|$ is vanishing by Lemma 1. Note that tighter bounds can be found about rows of $\boldsymbol{\Delta}$ similar to Lemma 12 in Dandi et al. (2023a), which is omitted here. Therefore, we focus on $\|\boldsymbol{f}_i^T(\boldsymbol{\Sigma}_c^{1/2} - \boldsymbol{I}_n)\|^2 = \boldsymbol{f}_i^T(\boldsymbol{\Sigma}_c^{1/2} - \boldsymbol{I}_n)^2 \boldsymbol{f}_i$ and utilize Hanson-Wright inequality.

For the last term, we reach the high probability event $|(\boldsymbol{f}_i^\perp)^T \boldsymbol{\Gamma}_c \boldsymbol{\kappa}_c| = \tilde{\mathcal{O}}(k^{\beta-1})$ as follows. Similar to the previous case, $\eta\boldsymbol{\Delta}_i$ does not affect our bound here. Therefore, we continue with $|\boldsymbol{f}_i^T \boldsymbol{\Gamma}_c \boldsymbol{\kappa}_c| = |\boldsymbol{f}_i^T \boldsymbol{\Gamma}_c(\boldsymbol{\Gamma}_c^T \boldsymbol{\Gamma}_c)^{-1}\boldsymbol{\Gamma}_c^T \boldsymbol{z}|$. Considering that the entries of $\boldsymbol{f}_i$ and the entries of $\boldsymbol{z}$ are sub-Gaussian, the product of any entry of $\boldsymbol{f}_i$ with any entry of $\boldsymbol{z}$ is sub-exponential with the sub-exponential norm of $Ck^{-1/2}$ for some $C > 0$. Then, $\boldsymbol{f}_i^T \boldsymbol{\Gamma}_c(\boldsymbol{\Gamma}_c^T \boldsymbol{\Gamma}_c)^{-1}\boldsymbol{\Gamma}_c^T \boldsymbol{z}$ term can be considered as a sum of zero-mean sub-exponential random variables. Therefore, we reach $|\boldsymbol{f}_i^T \boldsymbol{\Gamma}_c(\boldsymbol{\Gamma}_c^T \boldsymbol{\Gamma}_c)^{-1}\boldsymbol{\Gamma}_c^T \boldsymbol{z}| = \tilde{\mathcal{O}}(k^{-1/2})$ with high probability by Bernstein's inequality (Vershynin, 2018, Theorem 2.8.2).

Combining these together in (54), we reach $|a_{i|\kappa_c}| = \tilde{\mathcal{O}}(k^{\beta-1})$. Using $\epsilon = l(1-\beta) - 1 > 0$, we also get $|a_{i|\kappa_c}|^l = \tilde{\mathcal{O}}\left(k^{-(1+\epsilon)}\right)$ with high probability. $\qquad\square$

Since $|a_{i|\kappa_c}|^l = \tilde{\mathcal{O}}(1/k^{1+\epsilon})$ with high probability (Lemma 8), we decompose $\sigma(a_{i|\kappa_c} + b_i u_i)$ as

$$\sigma(a_{i|\kappa_c} + b_i u_i) = \sum_{o=0}^{l-1} \frac{1}{o!} \sigma^{(o)}(b_i u_i) a_{i|\kappa_c}^o + R_i, \tag{55}$$

$$= \sum_{o=0}^{l-1} \frac{1}{o!} \left( \sum_{j=0}^{\infty} \frac{1}{j!} h_{o+j} H_j((b_i/b)u_i) \right) a_{i|\kappa_c}^o + R_i, \tag{56}$$

$$= \sum_{j=0}^{\infty} \frac{1}{j!} \underbrace{\left( \sum_{o=0}^{l-1} \frac{1}{o!} h_{o+j} a_{i|\kappa_c}^o \right)}_{\hat{h}_j(a_{i|\kappa_c})} H_j((b_i/b)u_i) + R_i, \tag{57}$$

$$= \sum_{j=0}^{\infty} \frac{1}{j!} \hat{h}_j(a_{i|\kappa_c}) H_j((b_i/b)u_i) + R_i, \tag{58}$$

where $R_i = \tilde{\mathcal{O}}(1/k^{1+\epsilon})$ is the remainder since $|a_{i|\kappa_c}^l| = \tilde{\mathcal{O}}(1/k^{1+\epsilon})$ with $\epsilon > 0$. Here, we apply Taylor's expansion in the first step, Hermite expansion (10) in the second step, and basic algebraic manipulations for the rest of the steps.

Before continuing, it is beneficial to state the orthogonality of Hermite polynomials (Lemma 9) at this point, which is utilized in the following derivations of conditional mean and covariances of $\sigma(\cdot)$.

**Lemma 9** (Orthogonality of Hermite polynomials). *Let $H_i$ denote $i$-th probabilist's Hermite polynomial. If $(a, b)$ is jointly Gaussian with zero mean, then the following holds*

$$\mathbb{E}_{a,b}[H_i(a)H_j(b)] = i!(\mathbb{E}_{a,b}[ab])^i \delta_{ij}. \tag{59}$$

*Proof.* See O'Donnell (2014, Chapter 11.2) for the unit variance case ($\mathbb{E}[a^2] = \mathbb{E}[b^2] = 1$). For an extension that allows $\mathbb{E}[a^2] \neq 1$ while $\mathbb{E}[b^2] = 1$, see Demir & Dogan (2024). The same technique can be used for further extension to allow $\mathbb{E}[a^2] \neq 1$ and $\mathbb{E}[b^2] \neq 1$. $\qquad\square$

Now, we are in the position to study conditional mean and covariance of $\sigma(a_{i|\kappa_c} + b_i u_i)$. Note that $u_i$ is the source of randomness in the rest of the derivations while $a_{i|\kappa_c}$ and $b_i$ are fixed since we conditioned on $(c, \boldsymbol{\kappa}_c)$. First, we study the conditional mean as follows:

$$\nu_i(c, \boldsymbol{\kappa}_c) = \mathbb{E}_{u_i}[\sigma(a_{i|\kappa_c} + b_i u_i) \mid c, \boldsymbol{\kappa}_c], \tag{60}$$

$$= \sum_{j=0}^{\infty} \frac{1}{j!} \hat{h}_j(a_{i|\kappa_c}) \mathbb{E}_{u_i}[H_j((b_i/b)u_i)] + R_i, \tag{61}$$

$$= \hat{h}_0(a_{i|\kappa_c}) \mathbb{E}_{u_i}[H_0((b_i/b)u_i)] + R_i, \tag{62}$$

$$= \sum_{o=0}^{l-1} \frac{1}{o!} h_o a_{i|\kappa_c}^o + R_i, \tag{63}$$

$$= \mathbb{E}_{u_i}[\hat{\sigma}_l(a_{i|\kappa_c} + b_i u_i) \mid c, \boldsymbol{\kappa}_c] + R_i, \tag{64}$$

where $R_i = \tilde{\mathcal{O}}(1/k^{1+\epsilon})$ is the remainder which may vary line to line. Here, the first step is due to the decomposition in (58) while the second step follows from the orthogonality of Hermite polynomials (Lemma 9). The third step is due to $\mathbb{E}_{u_i}[H_0((b_i/b)u_i)] = 1$. Finally, the last step is because the $\hat{\sigma}_l$ function has the same first $l$ Hermite coefficients $h_j$ as those of the $\sigma$ function. Using (64), we then reach the following norm bound on the difference of the conditional means

$$\left\| \boldsymbol{\nu}(c, \boldsymbol{\kappa}_c) - \mathbb{E}[\hat{\sigma}_l(\hat{\boldsymbol{F}}\boldsymbol{x}) \mid c, \boldsymbol{\kappa}_c] \right\| \leq \sqrt{\sum_{i=1}^{k} R_i^2} = \tilde{\mathcal{O}}\left(\frac{1}{k^{1/2+\epsilon}}\right). \tag{65}$$

Next, we study the conditional cross-covariance between $\sigma(a_{i|\kappa_c} + b_i u_i)$ and $z_j^\perp$. Here, recall the relation between $u_i$ and $\boldsymbol{z}^\perp$ by definition: $u_i := (\boldsymbol{f}_i^\perp)^T \boldsymbol{z}^\perp / b_i$. Thus, $(u_i, z_j^\perp)$ is jointly Gaussian

with zero mean and $\mathbb{E}[u_i z_j^\perp] = f_{i,j}^\perp \mathbb{E}[(z_j^\perp)^2]/b_i$, which leads to the following derivation

$$\Psi_{i,j}(c, \boldsymbol{\kappa}_c) = \mathbb{E}[\sigma(a_{i|\kappa_c} + b_i u_i) z_j^\perp \mid c, \boldsymbol{\kappa}_c], \tag{66}$$

$$= \sum_{s=0}^{\infty} \frac{1}{s!} \hat{h}_s(a_{i|\kappa_c}) \mathbb{E}[H_s((b_i/b)u_i) z_j^\perp] + R_{i,j}, \tag{67}$$

$$= \hat{h}_1(a_{i|\kappa_c}) \mathbb{E}[H_1((b_i/b)u_i) z_j^\perp] + R_{i,j}, \tag{68}$$

$$= \sum_{o=0}^{l-1} \frac{1}{o!} h_{o+1} a_{i|\kappa_c}^o (b_i/b) \mathbb{E}[u_i z_j^\perp] + R_{i,j}, \tag{69}$$

$$= \sum_{o=0}^{l-1} \frac{1}{o!} h_{o+1} a_{i|\kappa_c}^o (f_{i,j}^\perp \mathbb{E}[(z_j^\perp)^2]/b) + R_{i,j}, \tag{70}$$

$$= \sum_{o=0}^{l-2} \frac{1}{o!} h_{o+1} a_{i|\kappa_c}^o (f_{i,j}^\perp \mathbb{E}[(z_j^\perp)^2]/b) + R_{i,j}, \tag{71}$$

$$= \mathbb{E}[\hat{\sigma}_l(a_{i|\kappa_c} + b_i u_i) z_j^\perp \mid c, \boldsymbol{\kappa}_c] + R_{i,j}, \tag{72}$$

where $R_{i,j} = \tilde{\mathcal{O}}(1/k^{1+\epsilon})$ is the remainder which can change line to line. Here, we start by applying the same steps we use for the derivation of (64). Then, we utilize $\mathbb{E}[u_i z_j^\perp] = f_{i,j}^\perp \mathbb{E}[(z_j^\perp)^2]/b_i$ to reach (70). For (71), we utilize the following high-probability event so that the $o = l - 1$ case is moved into the remainder $R_i$

$$\left| a_{i|\kappa_c}^{l-1} \right| \left| f_{i,j}^\perp \mathbb{E}[(z_j^\perp)^2]/b \right| = \tilde{\mathcal{O}}(1/k^{1+\epsilon}), \tag{73}$$

which follows from Lemma 8, $\mathbb{E}[(z_j^\perp)^2] = \tilde{\mathcal{O}}(1)$ by definition, $b \in \mathbb{R}^+$ by (A.8) and the following high-probability event: $|f_{i,j}^\perp| = \tilde{\mathcal{O}}(1/k^{1-\beta})$. Here, $|f_{i,j}^\perp| \le |f_{i,j}| + |\eta \Delta_{i,j}|$. Furthermore, $|f_{i,j}| = \tilde{\mathcal{O}}(1/k^{1/2})$ since $f_{i,j}$ is sub-Gaussian with sub-Gaussian norm of $C/\sqrt{\text{Tr}(\boldsymbol{\Sigma})}$ for some $C > 0$ by (A.6) and $\text{Tr}(\boldsymbol{\Sigma}) \asymp k$ by (A.4). Finally, $|\eta \Delta_{i,j}| \le \eta \|\Delta\| = \tilde{\mathcal{O}}(1/k^{1-\beta})$ by Lemma 1, which completes our explanation about (73).

To reach the last line (72) in our derivation of the cross-covariance, we use the fact that $\sigma$ and $\hat{\sigma}_l$ have the same first $l$ Hermite coefficients $h_j$. Using (72), we get

$$\left\| \boldsymbol{\Psi}(c, \boldsymbol{\kappa}_c) - \mathbb{E}[\hat{\sigma}_l(\hat{\boldsymbol{F}}\boldsymbol{x})(\boldsymbol{z}^\perp)^T \mid c, \boldsymbol{\kappa}_c] \right\| \le \left\| \boldsymbol{\Psi}(c, \boldsymbol{\kappa}_c) - \mathbb{E}[\hat{\sigma}_l(\hat{\boldsymbol{F}}\boldsymbol{x})(\boldsymbol{z}^\perp)^T \mid c, \boldsymbol{\kappa}_c] \right\|_F, \tag{74}$$

$$= \sqrt{\sum_{i=1}^{k} \sum_{j=1}^{n} R_{i,j}^2} = \tilde{\mathcal{O}}\left(\frac{1}{k^\epsilon}\right), \tag{75}$$

where we use the assumption that $n/k \in \mathbb{R}$ (A.1).

Before continuing with the covariance between $\sigma(a_{i|\kappa_c} + b_i u_i)$ and $\sigma(a_{j|\kappa_c} + b_j u_j)$, we have the following lemma about $\tilde{\boldsymbol{f}}_i^T \tilde{\boldsymbol{f}}_j$, which is helpful for the upcoming derivations on the covariance.

**Lemma 10.** *If $1/2 < \beta < 1$, then the following holds with high probability*

$$\max_{1 \le c \le \mathcal{C},\, 1 \le i,j \le k} \left| \tilde{\boldsymbol{f}}_i^T \tilde{\boldsymbol{f}}_j - b^2 \delta_{ij} \right| = \tilde{\mathcal{O}}(1/k^{1-\beta}), \tag{76}$$

*where $b := \sqrt{n/\text{Tr}(\boldsymbol{\Sigma})}$. Furthermore, $1 - \beta = (1 + \epsilon)/l$ by definition in Lemma 8.*

*Proof.* Recall that $\tilde{\boldsymbol{f}}_i := (\boldsymbol{I}_n - \boldsymbol{\Gamma}_c(\boldsymbol{\Gamma}_c^T \boldsymbol{\Gamma}_c)^{-1} \boldsymbol{\Gamma}_c^T) \boldsymbol{f}_i^\perp = (\boldsymbol{I}_n - \boldsymbol{\Gamma}_c(\boldsymbol{\Gamma}_c^T \boldsymbol{\Gamma}_c)^{-1} \boldsymbol{\Gamma}_c^T)(\boldsymbol{f}_i + \eta \boldsymbol{\Delta}_i)$. Then,

$$\left| \tilde{\boldsymbol{f}}_i^T \tilde{\boldsymbol{f}}_j - b^2 \delta_{ij} \right| = \left| (\boldsymbol{f}_i + \eta \boldsymbol{\Delta}_i)^T (\boldsymbol{I}_n - \boldsymbol{\Gamma}_c(\boldsymbol{\Gamma}_c^T \boldsymbol{\Gamma}_c)^{-1} \boldsymbol{\Gamma}_c^T)(\boldsymbol{f}_j + \eta \boldsymbol{\Delta}_j) - b^2 \delta_{ij} \right|, \tag{77}$$

$$\le \left| \boldsymbol{f}_i^T (\boldsymbol{I}_n - \boldsymbol{\Gamma}_c(\boldsymbol{\Gamma}_c^T \boldsymbol{\Gamma}_c)^{-1} \boldsymbol{\Gamma}_c^T) \boldsymbol{f}_j - b^2 \delta_{ij} \right| + \frac{\text{polylog } k}{k^{1-\beta}}, \tag{78}$$

$$\le |\boldsymbol{f}_i^T \boldsymbol{f}_j - b^2 \delta_{ij}| + |\boldsymbol{f}_i^T \boldsymbol{\Gamma}_c(\boldsymbol{\Gamma}_c^T \boldsymbol{\Gamma}_c)^{-1} \boldsymbol{\Gamma}_c^T \boldsymbol{f}_j| + \frac{\text{polylog } k}{k^{1-\beta}}, \tag{79}$$

$$\le \frac{\text{polylog } k}{k^{1-\beta}}, \tag{80}$$

where the first step is due to triangle inequality, $\eta\|\mathbf{\Delta}_i\| \leq \eta\|\mathbf{\Delta}\| = \tilde{\mathcal{O}}(k^{\beta-1})$ by Lemma 1 and $\|\mathbf{I}_n - \mathbf{\Gamma}_c(\mathbf{\Gamma}_c^T\mathbf{\Gamma}_c)^{-1}\mathbf{\Gamma}_c^T\| \leq 1 + \|\mathbf{\Gamma}_c(\mathbf{\Gamma}_c^T\mathbf{\Gamma}_c)^{-1}\mathbf{\Gamma}_c^T\| = \mathcal{O}(1)$ by definition and assumption (A.4). The second step follows from the triangle inequality. For the last step, we use the following two high-probability events: (i) $|\boldsymbol{f}_i^T\boldsymbol{f}_j - b^2\delta_{ij}| = \tilde{\mathcal{O}}\left(k^{-1/2}\right)$ and (ii) $|\boldsymbol{f}_i^T\mathbf{\Gamma}_c(\mathbf{\Gamma}_c^T\mathbf{\Gamma}_c)^{-1}\mathbf{\Gamma}_c^T\boldsymbol{f}_j| = \tilde{\mathcal{O}}\left(k^{-1/2}\right)$.

We prove (i) as follows. For $i \neq j$, we have $f_{i,s}$ and $f_{j,s}$ as sub-Gaussian random variables with sub-Gaussian norm bounded by $\frac{C}{\sqrt{\mathrm{Tr}(\mathbf{\Sigma})}}$ for some $C > 0$ due to Vershynin (2018, Example 2.5.8) and (A.6). Therefore, $f_{i,s}f_{j,s}$ is a sub-exponential random variable with sub-exponential norm bounded by $\frac{C^2}{\mathrm{Tr}(\mathbf{\Sigma})}$ due to Vershynin (2018, Lemma 2.7.7). Since $\boldsymbol{f}_i^T\boldsymbol{f}_j = \sum_{s=1}^n f_{i,s}f_{j,s}$, we can apply Bernstein's inequality (Vershynin, 2018, Theorem 2.8.2). The case of $i = j$ can be derived similarly. Note that $\mathrm{Tr}(\mathbf{\Sigma}) \asymp n$ by (A.4).

Similarly, we show (ii) in the following. The key point is that $\boldsymbol{f}_i$ is sampled independent of the fixed matrix $\mathbf{\Gamma}_c$ by (A.6). For $i = j$, we utilize Hanson-Wright inequality (Vershynin, 2018, Theorem 6.2.1) with $\mathbb{E}[\boldsymbol{f}_i^T\mathbf{\Gamma}_c(\mathbf{\Gamma}_c^T\mathbf{\Gamma}_c)^{-1}\mathbf{\Gamma}_c^T\boldsymbol{f}_i] = \mathrm{Tr}(\mathbf{\Gamma}_c(\mathbf{\Gamma}_c^T\mathbf{\Gamma}_c)^{-1}\mathbf{\Gamma}_c^T)/\mathrm{Tr}(\mathbf{\Sigma}) = \tilde{\mathcal{O}}(1/k)$ and $\|\mathbf{\Gamma}_c(\mathbf{\Gamma}_c^T\mathbf{\Gamma}_c)^{-1}\mathbf{\Gamma}_c^T\|_F = \tilde{\mathcal{O}}(1)$ by definition and (A.4). The case of $i \neq j$ follows similarly. $\square$

Then, we focus on the conditional covariance between $\sigma(a_{i|\kappa_c} + b_iu_i)$ and $\sigma(a_{j|\kappa_c} + b_ju_j)$. For $i \neq j$ (the off-diagonal entries of the covariance), we derive the covariance as follows:

$$\mathbb{E}[\sigma(a_{i|\kappa_c} + b_iu_i)\sigma(a_{j|\kappa_c} + b_ju_j) \mid c, \boldsymbol{\kappa}_c], \tag{81}$$

$$= \left(\sum_{s=0}^\infty \frac{1}{s!}\hat{h}_s(a_{i|\kappa_c})H_s((b_i/b)u_i) + R_i\right)\left(\sum_{s=0}^\infty \frac{1}{s!}\hat{h}_s(a_{j|\kappa_c})H_s((b_j/b)u_j) + R_j\right), \tag{82}$$

$$= \sum_{s=0}^\infty \frac{1}{(s!)^2}\hat{h}_s(a_{i|\kappa_c})\hat{h}_s(a_{j|\kappa_c})\mathbb{E}[H_s((b_i/b)u_i)H_s((b_j/b)u_j)] + R_{i,j}, \tag{83}$$

$$= \sum_{s=0}^\infty \frac{1}{s!}\hat{h}_s(a_{i|\kappa_c})\hat{h}_s(a_{j|\kappa_c})\left(\tilde{\boldsymbol{f}}_i^T\tilde{\boldsymbol{f}}_j/b^2\right)^s + R_{i,j}, \tag{84}$$

$$= \sum_{s=0}^{l-1}\frac{1}{s!}\left(\sum_{o=0}^{l-1}\frac{1}{o!}h_{s+o}a_{i|\kappa_c}^o\right)\left(\sum_{o=0}^{l-1}\frac{1}{o!}h_{s+o}a_{j|\kappa_c}^o\right)\left(\tilde{\boldsymbol{f}}_i^T\tilde{\boldsymbol{f}}_j/b^2\right)^s + R_{i,j}, \tag{85}$$

$$= \sum_{s=0}^{l-1}\frac{1}{s!}\left(\sum_{o=0}^{l-1-s}\frac{1}{o!}h_{s+o}a_{i|\kappa_c}^o\right)\left(\sum_{o=0}^{l-1-s}\frac{1}{o!}h_{s+o}a_{j|\kappa_c}^o\right)\left(\tilde{\boldsymbol{f}}_i^T\tilde{\boldsymbol{f}}_j/b^2\right)^s + R_{i,j}, \tag{86}$$

$$= \mathbb{E}[\hat{\sigma}_l(a_{i|\kappa_c} + b_iu_i)\hat{\sigma}_l(a_{j|\kappa_c} + b_ju_j) \mid c, \boldsymbol{\kappa}_c] + R_{i,j}, \tag{87}$$

where $R_i, R_{i,j} = \tilde{\mathcal{O}}(1/k^{1+\epsilon})$ are the remainder terms that may vary from line to line. For the first step, we use the decomposition in (58) while the second and third steps are due to the orthogonality of Hermite polynomials (Lemma 9) and the fact that $(u_i, u_j)$ is jointly Gaussian with $\mathbb{E}[u_iu_j] = \tilde{\boldsymbol{f}}_i^T\tilde{\boldsymbol{f}}_j/(b_ib_j)$. To reach (85), we use the high-probability event: $|\tilde{\boldsymbol{f}}_i^T\tilde{\boldsymbol{f}}_j/b^2|^l = \tilde{\mathcal{O}}(1/k^{1+\epsilon})$, which follows from Lemma 10. To arrive at (86), we utilize the following high-probability event

$$\left|a_{i|\kappa_c}\right|^o\left|\tilde{\boldsymbol{f}}_i^T\tilde{\boldsymbol{f}}_j/b^2\right|^s = \tilde{\mathcal{O}}(1/k^{1+\epsilon}), \quad \text{for} \quad o + s \geq l, \tag{88}$$

which again follows from Lemma 8, Lemma 10 and $b \in \mathbb{R}^+$ by (A.8). Finally, the last line (87) is due to the fact that $\hat{\sigma}_l$ function has the same first $l$ Hermite coefficients $h_j$ as those of $\sigma$ function.

For $i = j$ (the diagonal entries of the covariance), we apply Taylor's expansion to $\sigma(a_{i|\kappa_c} + b_iu_i)^2$ around $bu_i$ as follows:

$$\sigma(a_{i|\kappa_c} + b_iu_i)^2 = \sigma(bu_i)^2 + R_{i,i}, \tag{89}$$

where $R_{i,i} := 2\sigma(t)\sigma'(t)(a_{i|\kappa_c} + (b_i - b)u_i)$ for some $t$ in-between $bu_i$ and $a_{i|\kappa_c} + b_iu_i$. Here, we see that $|t| = \tilde{\mathcal{O}}(1)$ with high probability by definition, which makes $|\sigma(t)\sigma'(t)| = \tilde{\mathcal{O}}(1)$ with high probability since $\sigma$ has bounded derivatives by Assumption (A.8). Then, since $|a_{i|\kappa_c}| = \tilde{\mathcal{O}}(1/k^{(1+\epsilon)/l})$

by Lemma 8 and $|b_i - b| = \tilde{\mathcal{O}}(1/k^{(1+\epsilon)/l})$ by Lemma 10, and $u_i \sim \mathcal{N}(0, 1)$, we can conclude that $R_{i,i} = \tilde{\mathcal{O}}(1/k^{(1+\epsilon)/l})$ holds with high probability. Using this result, we can reach

$$\mathbb{E}[\sigma(a_{i|\kappa_c} + b_i u_i)^2 \mid c, \boldsymbol{\kappa}_c] = \mathbb{E}[\sigma(b u_i)^2 \mid c, \boldsymbol{\kappa}_c] + R_{i,i}, \tag{90}$$

$$= \mathbb{E}[\hat{\sigma}_l(a_{i|\kappa_c} + b_i u_i)^2 \mid c, \boldsymbol{\kappa}_c] + R_{i,i}, \tag{91}$$

where $R_{i,i} = \tilde{\mathcal{O}}(1/k^{(1+\epsilon)/l})$ is again the remainder that can differ from line to line. To get the final line, we use $\mathbb{E}[\sigma(b u_i)^2] = \mathbb{E}[\hat{\sigma}_l(b u_i)^2] = (h_l^*)^2 + \sum_{j=0}^{l-1} h_j^2/(j!)$ by the definition of $\hat{\sigma}_l$ (16).

Therefore, we reach the following norm bound on the difference of the covariances for $\sigma$ and $\hat{\sigma}_l$

$$\left\| \mathbb{E}[\sigma(\hat{\boldsymbol{F}}\boldsymbol{x})\sigma(\hat{\boldsymbol{F}}\boldsymbol{x})^T \mid c, \boldsymbol{\kappa}_c] - \mathbb{E}[\hat{\sigma}_l(\hat{\boldsymbol{F}}\boldsymbol{x})\hat{\sigma}_l(\hat{\boldsymbol{F}}\boldsymbol{x})^T \mid c, \boldsymbol{\kappa}_c] \right\| \leq \max_i |R_{i,i}| + \sqrt{\sum_{i \neq j} R_{i,j}^2}, \tag{92}$$

$$= \tilde{\mathcal{O}}\left(1/k^{\min(\epsilon, (1+\epsilon)/l)}\right), \tag{93}$$

where we first separate diagonal and off-diagonal parts in the first line while the last line follows from (91) and (87). This completes our task of showing the conditional mean and covariances of $\hat{\sigma}_l(\hat{\boldsymbol{F}}\boldsymbol{x})$ are equivalent to those of $\sigma(\hat{\boldsymbol{F}}\boldsymbol{x})$ when conditioned on $(c, \boldsymbol{\kappa}_c)$.

**Conditional Gaussian Equivalence under Equivalent Means and Covariances**
From the previous derivations, we have

$$\left\| \mathbb{E}\left[ \sigma(\hat{\boldsymbol{F}}\boldsymbol{x}) \mid c, \boldsymbol{\kappa}_c \right] - \mathbb{E}\left[ \hat{\sigma}_l(\hat{\boldsymbol{F}}\boldsymbol{x}) \mid c, \boldsymbol{\kappa}_c \right] \right\| = o(1), \tag{94}$$

$$\left\| \mathbb{E}\left[ \sigma(\hat{\boldsymbol{F}}\boldsymbol{x})(\boldsymbol{z}^\perp)^T \mid c, \boldsymbol{\kappa}_c \right] - \mathbb{E}\left[ \hat{\sigma}_l(\hat{\boldsymbol{F}}\boldsymbol{x})(\boldsymbol{z}^\perp)^T \mid c, \boldsymbol{\kappa}_c \right] \right\|_F = o(1), \tag{95}$$

$$\left\| \text{Cov}\left( \sigma(\hat{\boldsymbol{F}}\boldsymbol{x}) \mid c, \boldsymbol{\kappa}_c \right) - \text{Cov}\left( \hat{\sigma}_l(\hat{\boldsymbol{F}}\boldsymbol{x}) \mid c, \boldsymbol{\kappa}_c \right) \right\| = o(1), \tag{96}$$

from (65), (75) and (93), respectively, under the conditions of Theorem 4.

In the rest of this section, we describe why (94)-(96) are enough to utilize the conditional Gaussian equivalence (Theorem 3) for the proof of Theorem 4. First of all, note that we only use the exact mean and exact covariances in the proof of the conditional CLT (Lemma 5) while exact equality conditions can be directly replaced with (94)-(96) for the rest of our proof for Theorem 3.

First, we observe that (94) and (95) do not cause any significant change in comparison to the exact equality since $\psi$ in Lemma 5 is Lipschitz function. To show this, we define the following remainder

$$\boldsymbol{r}(c, \boldsymbol{\kappa}_c) := (\hat{\boldsymbol{\nu}}(c, \boldsymbol{\kappa}_c) - \boldsymbol{\nu}(c, \boldsymbol{\kappa}_c)) + (\hat{\boldsymbol{\Psi}}(c, \boldsymbol{\kappa}_c) - \boldsymbol{\Psi}(c, \boldsymbol{\kappa}_c))\boldsymbol{z}^\perp \tag{97}$$

where $\hat{\boldsymbol{\nu}}(c, \boldsymbol{\kappa}_c)$ and $\hat{\boldsymbol{\Psi}}(c, \boldsymbol{\kappa}_c)$ are defined as

$$\hat{\boldsymbol{\nu}}(c, \boldsymbol{\kappa}_c) := \mathbb{E}\left[ \hat{\sigma}_l(\hat{\boldsymbol{F}}\boldsymbol{x}) \mid c, \boldsymbol{\kappa}_c \right], \qquad \hat{\boldsymbol{\Psi}}(c, \boldsymbol{\kappa}_c) := \mathbb{E}\left[ \hat{\sigma}_l(\hat{\boldsymbol{F}}\boldsymbol{x})(\boldsymbol{z}^\perp)^T \mid c, \boldsymbol{\kappa}_c \right]. \tag{98}$$

Then, we have the following about the test function $\psi$ in Lemma 5

$$\left| \psi\left( \tilde{\boldsymbol{w}}^T \left( \hat{\phi}(\boldsymbol{x}) + \boldsymbol{r}(c, \boldsymbol{\kappa}_c) \right), \boldsymbol{\xi}^T \boldsymbol{x} \right) - \psi\left( \tilde{\boldsymbol{w}}^T \hat{\phi}(\boldsymbol{x}), \boldsymbol{\xi}^T \boldsymbol{x} \right) \right| \leq L \left| \tilde{\boldsymbol{w}}^T \boldsymbol{r}(c, \boldsymbol{\kappa}_c) \right|, \tag{99}$$

for some $L > 0$, which follows from the fact that $\psi$ is a Lipschitz function. Since $\tilde{\boldsymbol{w}} \in \mathcal{S}_k$ in Lemma 5, we have $\|\tilde{\boldsymbol{w}}\| = \mathcal{O}(1)$. Thus, we just need to show that $\|\boldsymbol{r}(c, \boldsymbol{\kappa}_c)\|$ is vanishing, which would then imply that replacing $\hat{\phi}(\boldsymbol{x})$ with $\hat{\phi}(\boldsymbol{x}) + \boldsymbol{r}(c, \boldsymbol{\kappa}_c)$ has a vanishing effect on the test function (99). To bound the norm of $\|\boldsymbol{r}(c, \boldsymbol{\kappa}_c)\|$, we first apply the triangle inequality as follows:

$$\|\boldsymbol{r}(c, \boldsymbol{\kappa}_c)\| \leq \|\hat{\boldsymbol{\nu}}(c, \boldsymbol{\kappa}_c) - \boldsymbol{\nu}(c, \boldsymbol{\kappa}_c)\| + \|(\hat{\boldsymbol{\Psi}}(c, \boldsymbol{\kappa}_c) - \boldsymbol{\Psi}(c, \boldsymbol{\kappa}_c))\boldsymbol{z}^\perp\|, \tag{100}$$

where the first term is vanishing by (94) while the second term deserves further elaboration. Here, the second term has a vanishing bound with high probability by Hanson-Wright inequality (Vershynin, 2018, Theorem 6.2.1) since $\boldsymbol{z}^\perp$ is a random vector with independent mean zero sub-Gaussian coordinates and we have (95). This completes our explanation about the impact of (94) and (95).

Now, we focus on the covariance (96). At this point, recall that our proof of Lemma 5 reduces to Theorem 2 and Lemma 2 in Hu & Lu (2023). To cover the effect of (96), we replace Lemma 5 in Hu & Lu (2023) with our bound (96) for the covariance. Note that the vanishing bound in (96) is enough to replace the bound in Hu & Lu (2023) for our case since we just need a vanishing bound in their Lemma 2 and Theorem 2. For the sake of completeness, we would like to mention the location where the bound for the covariance is utilized in Hu & Lu (2023): it is applied just above equation (79) in the proof of their Lemma 2, which then resulted in equation (68) in their Lemma 2 and equation (67) in their Theorem 2 after some derivation.

We showed that the conditional Gaussian equivalence in Theorem 3 is applicable with (94)-(96). Therefore, we utilize Theorem 3 with (94)-(96), which completes our proof for Theorem 4.

## D   ADDITIONAL SIMULATION RESULTS

Here, we provide additional simulation results about (i) landscape of generalization error with respect to $\alpha$ and $\beta$ (Figure 4), (ii) impact of $\beta$ for different $\alpha$ values (Figure 5), (iii) training errors (Figure 6) and (iv) the effect learning rate for Fashion-MNIST classification (Figure 7). These are beneficial for understanding various aspects of our setting that we could not cover in the main text due to page limitations.

### D.1   HEAT MAPS OF GENERALIZATION ERRORS WITH RESPECT TO $\alpha$ AND $\beta$

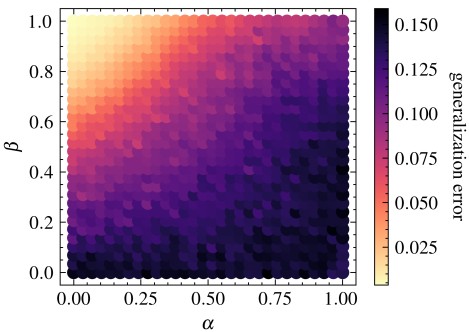
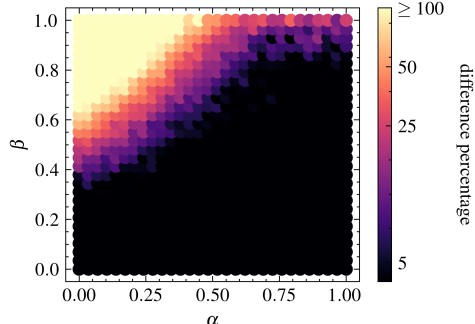

(a) Generalization error of the neural network. The data spread scales as $\|\mathbf{\Sigma}\| \asymp n^{\beta(1-\alpha)}$, while the learning rate scales as $\eta \asymp n^{\beta\alpha}$.

(b) Percentage difference between the generalization errors of the neural network and the equivalent linear model (8).

Figure 4: Generalization performance of the neural network with respect to $\alpha$ and $\beta$ as heat maps. We set $n = 400, m = 500$, and used the ReLU activation function ($\sigma = \sigma_* = $ ReLU). The number of classes is $\mathcal{C} = 2$ with dimensions $d_1 = d_2 = 1$. The parameters $\theta_{1,1} = \theta_{2,1} = n^{\beta(1-\alpha)}$ and $\lambda = 1e-4$ are employed to control the model's behavior. $\boldsymbol{\xi} = (\boldsymbol{\gamma}_{1,1} + \boldsymbol{\gamma}_{2,1})/(\|\boldsymbol{\gamma}_{1,1} + \boldsymbol{\gamma}_{2,1}\|\|\mathbf{\Sigma}^{1/2}\|)$ is used to ensure high alignment between $\boldsymbol{\xi}$ and the data covariance. The results presented are averages from 20 Monte Carlo simulations.

Figure 4a illustrates the overall landscape of the generalization error with respect to $\alpha$ and $\beta$, highlighting the significance of the strength parameter $\beta$, which governs the scale of the learning rate $\eta$ combined with the data spread $\|\mathbf{\Sigma}\|$. Our findings indicate that as we increase $\beta$ and/or decrease $\alpha$, generalization performance improves, underscoring the importance of structured data relative to first-layer learning. Additionally, Figure 4b demonstrates that surpassing linear models' performance—an important benchmark in the literature (Cui et al., 2024)—requires increasing $\beta$ while sometimes only reducing $\alpha$ suffices within certain ranges ($\beta \in [0.4, 0.7]$). Overall, these observations highlight the profound impact of the structure of data on learning outcomes.

## D.2 IMPACT OF $\beta$ FOR DIFFERENT $\alpha$ VALUES

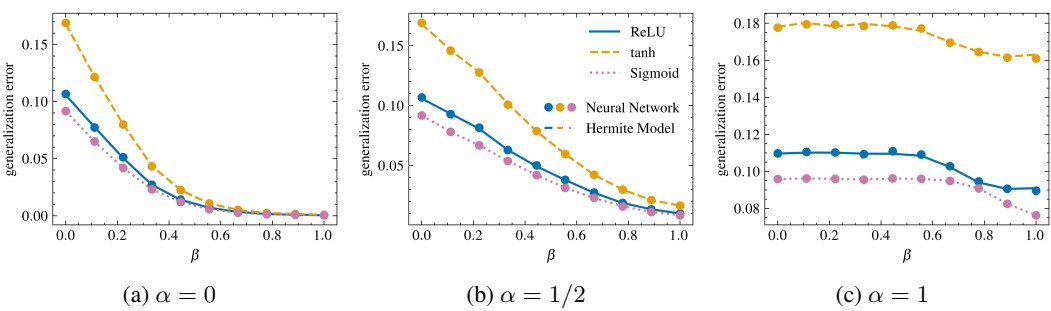

(a) $\alpha = 0$      (b) $\alpha = 1/2$      (c) $\alpha = 1$

Figure 5: Impact of $\beta$ for various $\alpha$ values in the setting of Figure 1c.

In Figure 5, we observe how the generalization curve with respect to $\beta$ gets affected by $\alpha$. Overall, we see that an increase in $\beta$ is beneficial for improving the generalization error, while the $\alpha$ value shapes the generalization curve with respect to $\beta$.

## D.3 TRAINING ERRORS

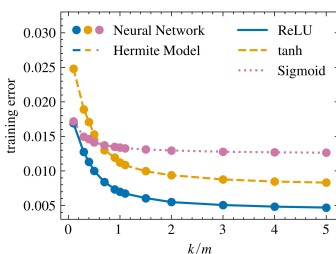

Figure 6: Equivalence of the training errors with respect to $k/m$ in the setting of Figure 1a.

In the main text, we focus on the generalization error for our simulation results since it is of more interest in comparison to the training error. However, it is important to note that our results hold for the training error as well, which may be of particular interest on its own. Therefore, here, we provide a simulation result about the training errors in Figure 6. We observe that the training error decreases as $k/m$ (governing the number of hidden neurons) increases, which is expected. More significantly, we see that the training errors of two-layer neural networks match those of the equivalent Hermite model for all activation functions in the simulation.

## D.4 EFFECT OF LEARNING RATE BEYOND $\beta \in [0, 1]$ FOR FASHION-MNIST CLASSIFICATION

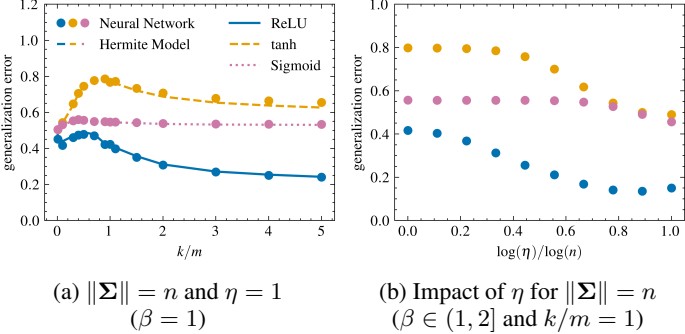

(a) $\|\mathbf{\Sigma}\| = n$ and $\eta = 1$      (b) Impact of $\eta$ for $\|\mathbf{\Sigma}\| = n$
         ($\beta = 1$)          ($\beta \in (1, 2]$ and $k/m = 1$)

Figure 7: Impact of the learning rate for Fashion-MNIST classification in the setting of Figure 3a.

While our result in Figure 3 for Fashion-MNIST classification is interesting on its own, one can question the impact of the learning rate $\eta$ in the setting of the figure. Here, we answer the aforementioned question with a new simulation result in Figure 7. Note that Figure 7a is the same as Figure 3a, which is given again for the sake of side-to-side comparison with 7b. In Figure 7b, we can see that the generalization performance gets better as we approach towards $\eta \asymp n^1$, which aligns with the predictions for the case of isotropic data. However, we lose the equivalence since $\beta > 1$. Thus, the corresponding Hermite model is dropped from the plot. To study the result in (b), one needs to extend our main results such that they cover $\beta \in (1, 2]$, which is left to future work.

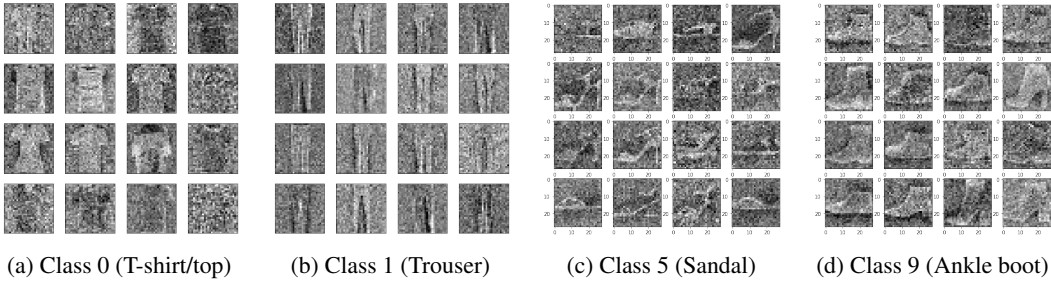

(a) Class 0 (T-shirt/top)      (b) Class 1 (Trouser)      (c) Class 5 (Sandal)      (d) Class 9 (Ankle boot)

Figure 8: Examples of input images after the pre-processing for the result in Figure 3.

## E    DETAILS FOR THE FASHION-MNIST SIMULATIONS

In this section, we detail our numerical simulation for Fashion-MNIST classification. We begin by training a conditional Generative Adversarial Network (cGAN) using the Fashion-MNIST dataset, following the implementation of InfoGAN (Chen et al., 2016) as outlined by Dandi et al. (2023b). The cGAN is trained for 50 epochs with a batch size of 64, utilizing the Adam optimizer with a learning rate of 0.0002 and $\beta$ parameters set to (0.5, 0.999). The model is optimized using binary cross-entropy (BCE) loss for both the generator and discriminator. This structured approach allows us to effectively generate class-conditioned samples, facilitating robust classification tasks while adhering to established training protocols.

Next, we preprocess the input images generated by the trained cGAN model to align with our assumptions outlined in (A.2)-(A.4). Each sample is flattened into a vector in $\mathbb{R}^n$, where $n = 28 \times 28 = 784$. Before preprocessing, we calculate the mean and covariance for each class using Monte Carlo estimation with 1 million samples per class. We then compute the ratio $t^2 := \text{Tr}(\boldsymbol{\Sigma}_1)/\text{Tr}(\boldsymbol{\Sigma}_2)$ to adjust the scale of the second class samples, ensuring that $\text{Tr}(\boldsymbol{\Sigma}_1) = \text{Tr}(\boldsymbol{\Sigma}_2)$ as required in (A.3). The preprocessing involves demeaning samples from both classes and scaling the second class by $t$, resulting in $\boldsymbol{\mu}_1 = \boldsymbol{\mu}_2 = \boldsymbol{0}$ and $\text{Tr}(\boldsymbol{\Sigma}_1) = \text{Tr}(\boldsymbol{\Sigma}_2)$. To further satisfy assumption (A.4), we introduce noise to the samples using the formula $\boldsymbol{x} := \sqrt{n/\text{Tr}(\boldsymbol{\Sigma}_1)}\bar{\boldsymbol{x}} + \boldsymbol{\epsilon}$, where $\boldsymbol{\epsilon} \sim \mathcal{N}(0, \boldsymbol{I}_n)$ and $\bar{\boldsymbol{x}}$ denotes the preprocessed sample. The multiplier $\sqrt{n/\text{Tr}(\boldsymbol{\Sigma}_1)}$ controls the signal-to-noise ratio (SNR), consistent with our data assumptions. Labels are assigned as $+1$ for the first class and $-1$ for the second. This completes our preprocessing steps, and sample images generated through this procedure are shown in Figure 8, illustrating clear distinguishability between classes. Additionally, we use $b = 1$ for calculating the Hermite coefficients in our Fashion-MNIST classification simulations.

## F    EXTENSION TO NON-ZERO MEANS FOR MIXTURE COMPONENTS

We assume zero mean for the mixture components ($\boldsymbol{\mu}_c = \boldsymbol{0}$) in Assumption (A.3) to simplify our proofs. However, it is important to recognize that the mean $\boldsymbol{\mu}_c$ of each Gaussian component functions similarly to the spikes $\boldsymbol{\gamma}_{c,i}$ in the covariance structure described by equation (9). To extend our analysis to include non-zero means, we assume the existence of a constant $C > 0$ such that $\|\boldsymbol{\mu}_c\|^2 \leq C\|\boldsymbol{\Sigma}\|$. This allows us to incorporate non-zero means $\boldsymbol{\mu}_c \neq \boldsymbol{0}$ by adding the term $\hat{\mathbf{F}}\boldsymbol{\mu}_c$ into the structure outlined in Lemma 2.

In our proofs, we would need to bound additional terms arising from these non-zero means; however, we omit these details here for brevity. Instead, we present numerical evidence demonstrating that our

results remain valid even when accounting for non-zero means. Figure 9 illustrates our numerical simulations for Fashion-MNIST classification with non-zero means ($\boldsymbol{\mu}_c \neq \boldsymbol{0}$). The results indicate that the Hermite model performs equivalently to the neural networks, even with non-zero means for the inputs of each class. Notably, the generalization errors in the case with non-zero means (as shown in Figure 9) are lower compared to those observed with zero means (Figure 3), suggesting that classification becomes easier when different means are introduced. This finding further underscores the robustness of our framework and its applicability to more complex data scenarios.

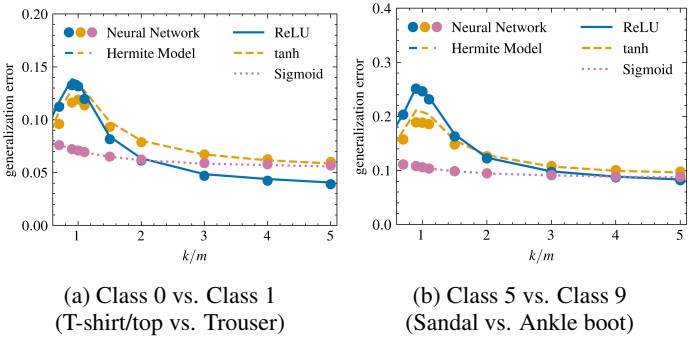

(a) Class 0 vs. Class 1
(T-shirt/top vs. Trouser)

(b) Class 5 vs. Class 9
(Sandal vs. Ankle boot)

Figure 9: Impact of non-zero means on Fashion-MNIST classification for $\|\boldsymbol{\Sigma}\| = n$ and $\eta = 1$. $m = 500, \lambda = 1e - 5$, and $l = 5$. This is the same setting as Figure 3, however inputs for each class are not demeaned separately; instead, all data is demeaned together to ensure $\mathbb{E}[\boldsymbol{x}] = \boldsymbol{0}$ while maintaining $\boldsymbol{\mu}_c \neq \boldsymbol{0}$. Comparison with results from Figure 3 highlights the effects of non-zero means on generalization error.

