# OpenReview forum: "Asymptotic Analysis of Two-Layer Neural Networks after One Gradient Step under Gaussian Mixtures Data with Structure"
_ICLR.cc/2025/Conference — ICLR 2025 Poster_

### Official Review · Reviewer_Veqf · 2024-10-21

**Soundness:** 4
**Presentation:** 4
**Contribution:** 2
**Rating:** 6
**Confidence:** 3

**Summary:**

The authors consider the problem of characterizing the hidden-layer features of a two layer neural network, after one gradient step on the first layer weights, in the asymptotic limit of comparably large dimension, width and number of samples. They consider (mixtures of) Gaussians with spiked covariance structure. They establish a conditional Gaussian equivalent map for the features, and further establish that the activation function can be replaced by a polynomial function with asymptotically equivalent test and train errors. Different aspects of the model (learning rate -to- data spread ratio, alignment between the spikes, cluster weights) are varied and their effect on the test error discussed. Many numerical experiments are performed to support or illustrate the discussion.

**Strengths:**

The paper is well written; results are sufficiently discussed and put in context. Section 6 provides ample intuition and discussion on the effect of the different parameters of the problem, allowing to illustrate and support the experimental results. Although I did not carefully check the proof, technical results seem sound.

Technically, the analysis of conditional Gaussian universality to structured covariances is interesting and relevant to extend the stream of recent related works to more realistic data distributions. In particular, the analysis of the joint scaling of $\eta, \lVert \Sigma\lVert$ is interesting, and provides a insightful picture of the tradeoff. I appreciated section 6, and in particular the discussion of the effect of the alignment between different covariance spikes.

**Weaknesses:**

I do have a number of clarifying questions, which I list in the section below. Overall, I believe the work is sound and will be of interest to the community working on those topics. I do not give a higher score, because the paper leaves open rather natural questions which are naturally attached to its setting, beyond its current scope (namely establish Gaussian universality for the considered distribution). In particular :

- Can a description of the features spectrum be given in the case of a single-index target function, in the likeliness of Moniri et al. (2023)? Do the spikes persist, or are they altered by the presence of the structure in the covariances ?

- Is it possible to reach a tight asymptotic expression for the test and train errors?

In regards to this, I believe the title would gain to be slightly toned down slightly to be better representative of the scope of this paper, which the abstract accurately on the other hand reflects. I still stress however that the contribution is interesting and sound.

**Questions:**

-  In (17), does $\hat{F}$ still refer to (12), namely the first layer weights after a gradient step on the _original_ network with activation $\sigma$, and not $\hat{\sigma}_l$? In other words, does the $\sigma\leftrightarrow \hat{\sigma}_l$ equivalence hold only when training the readout, or does it hold for the full training procedure, i.e. including when performing the gradient step on the first layer? This is not clear from the discussion in the main text.

- Can the authors add a discussion of why considering zero means (assumption A.3) is necessary, or if it can be relaxed to accommodate non-zero means ?

- l. 410 : "This equivalence significantly simplifies our analysis by transforming the nonlinear activation into a polynomial form, thereby enhancing the tractability of the model while maintaining its performance characteristics." Can the authors be more specific on which point of the analysis this equivalence aids? This point, if any, is not sufficiently discussed in the current state of the manuscript.

---

> ### Author Response · Authors · 2024-11-21
> **Part 1/2**
>
> Thank you for your insightful questions and feedback regarding our paper. We appreciate your recognition of the contribution's soundness and relevance to the community. Below, we address your specific concerns regarding the characterization of the feature spectrum and the derivation of tight asymptotic expressions for test and train errors.
>
> > I do have a number of clarifying questions, which I list in the section below. Overall, I believe the work is sound and will be of interest to the community working on those topics. I do not give a higher score, because the paper leaves open rather natural questions which are naturally attached to its setting, beyond its current scope (namely establish Gaussian universality for the considered distribution). In particular :
>
> > Can a description of the features spectrum be given in the case of a single-index target function, in the likeliness of Moniri et al. (2023)? Do the spikes persist, or are they altered by the presence of the structure in the covariances ?
>
> We appreciate the opportunity to address the characterization of the feature spectrum in relation to a single-index target function, as discussed in Moniri et al. (2023). Our research indeed facilitates a comprehensive characterization of the feature spectrum. Specifically, we can leverage Lemma 2, which presents the Bulk + Structure decomposition of $\hat{\mathbf{F}} \mathbf{x}$, alongside Theorem 4, which pertains to equivalent polynomial activation. This combination enables us to characterize the feature spectrum in a manner analogous to that outlined by Moniri et al. (2023). In our framework, the number of spikes within the feature spectrum is influenced by the strength parameter $\beta$, which governs both the learning rate and data spread. This contrasts with Moniri et al. (2023), where the spike count is solely dependent on the learning rate. It is important to note that while our model provides insights into the general behavior of spikes, pinpointing their exact characterization necessitates further investigation.
>
> > Is it possible to reach a tight asymptotic expression for the test and train errors?
>
> Thank you for your insightful question regarding the possibility of deriving tight asymptotic expressions for test and train errors. We believe that our results provide a solid foundation for such an analysis. By leveraging our findings related to Gaussian equivalence, particularly concerning the means and covariances of features for each class, we can apply established performance characterization techniques. Specifically, we can draw upon results achieved through the replica method [Loureiro et al., 2021] or random matrix theory [Dandi et al., 2024]. These approaches hold significant promise for achieving tight asymptotic expressions for both test and train errors. However, it is important to note that while these avenues are indeed fruitful, they extend beyond the current scope of our submission, as indicated by the reviewer. We appreciate the opportunity to discuss this potential direction for future research and look forward to further exploring these methodologies in subsequent work.
>
> [Loureiro et al., 2021)]: Learning gaussian mixtures with generalized linear models: Precise asymptotics in high-dimensions (NeurIPS).
>
> [Dandi et al., 2024]: A Random Matrix Theory Perspective on the Spectrum of Learned Features and Asymptotic Generalization Capabilities (arXiv:2410.18938).
>
> > In regards to this, I believe the title would gain to be slightly toned down slightly to be better representative of the scope of this paper, which the abstract accurately on the other hand reflects. I still stress however that the contribution is interesting and sound.
>
> Thank you for your thoughtful feedback regarding the title of our paper. We appreciate your suggestion to better align it with the scope of the work. We chose the phrase "Asymptotic Analysis" to emphasize the asymptotic and theoretical nature of our research. However, we understand the importance of ensuring that the title accurately reflects the content and contributions of the paper. Based on your feedback, we will certainly consider adjusting the title to enhance its alignment with the paper's scope, if permitted. We are glad to hear that you find our contributions interesting and sound, and we value your input as we strive for clarity and precision in our presentation.

---

> > ### Author Response · Authors · 2024-11-21
> > **Part 2/2**
> >
> > > Questions:
> >
> > > In (17), does $\hat{\mathbf{F}}$ still refer to (12), namely the first layer weights after a gradient step on the original network with activation
> >  $\sigma$, and not $\hat{\sigma}_l$? In other words, does the $\sigma \leftrightarrow \hat{\sigma}_l$ equivalence hold only when training the readout, or does it hold for the full training procedure, i.e. including when performing the gradient step on the first layer? This is not clear from the discussion in the main text.
> >
> >  Thank you for highlighting the point of confusion regarding equation (17). To clarify, in this context, $\hat{\mathbf{F}}$ indeed refers to the "trained" weight matrix of the first layer, as defined in equation (3). Our intention is to demonstrate the equivalence that arises after the training of this first layer, which is crucial for understanding the subsequent analysis. We clarified this point in the revised paper to ensure that readers can easily grasp this important aspect of our work.
> >
> > > Can the authors add a discussion of why considering zero means (assumption A.3) is necessary, or if it can be relaxed to accommodate non-zero means ?
> >
> > Thank you for your insightful question regarding the necessity of assuming zero means in our analysis (Assumption A.3). We appreciate the opportunity to clarify this point. While the mean $\boldsymbol{\mu}_c$ of each Gaussian component does indeed function similarly to spikes within the covariance structure $\boldsymbol{\Sigma}_c$ —as noted by reviewer rEU8— we initially adopted the zero mean assumption to streamline our proofs and focus on the core aspects of our model. However, in response to reviewer feedback, we have expanded our discussion in the revised paper to address this issue comprehensively. In Appendix F, we now include a detailed analysis of how the zero mean assumption can be relaxed to accommodate non-zero means. This section not only outlines the theoretical implications of such a relaxation but also presents a new result that provides numerical evidence demonstrating that our main claims remain valid even when $\boldsymbol{\mu}_c \neq \mathbf{0}$. We believe these additions will enhance the clarity and applicability of our findings, and we appreciate your suggestion to further elaborate on this critical aspect.
> >
> > > l. 410 : "This equivalence significantly simplifies our analysis by transforming the nonlinear activation into a polynomial form, thereby enhancing the tractability of the model while maintaining its performance characteristics." Can the authors be more specific on which point of the analysis this equivalence aids? This point, if any, is not sufficiently discussed in the current state of the manuscript.
> >
> > Thank you for your important question regarding the specific points in our analysis where the equivalence of nonlinear activations to polynomial forms aids our work. The transformation to a finite-degree polynomial activation significantly enhances the tractability of our model, particularly in performance characterization. Polynomial activations, as opposed to general nonlinear functions like ReLU or tanh, allow for a more straightforward study of the model's behavior and performance metrics. For instance, Theorem 4 establishes an equivalence class of activation functions, indicating that those with identical Hermite coefficients yield the same training and generalization performance. This insight not only simplifies our analysis but also opens avenues for optimizing activation functions based on their polynomial representations. By characterizing these polynomial forms, we can identify "optimal" activations that maximize performance. We expanded on these points in the revised manuscript to provide clearer connections between the equivalence and its implications for our analysis.

---

### Official Review · Reviewer_rEU8 · 2024-10-28

**Soundness:** 3
**Presentation:** 3
**Contribution:** 2
**Rating:** 6
**Confidence:** 3

**Summary:**

This manuscript investigates the training and generalization performance of two-layer neural networks after a single gradient descent step, under a structured covariate model given by a Gaussian mixture with finite-rank + identity covariance. The analysis is conducted in the proportional high-dimensional limit where the covariate dimension, hidden-layer width, and sample size diverge at constant rate. Leveraging recent progress in the Gaussian universality literature, the authors show that:

- After a single-gradient step on the first layer weights, the gradient admits a decomposition in terms of a rank-1 spike and a correction which is asymptotically small in operator norm.
- A conditional Gaussian equivalence theorem showing that the training and generalization errors after a single-gradient step are asymptotically equivalent to conditional Gaussian covariate model. This equivalence can be further written in terms of a Hermite decomposition of the non-linear activation.

Finally, the authors provide empirical validation via simulations on Fashion-MNIST classification tasks, supporting the theoretical predictions and highlighting how data structure influences learning outcomes.

**Strengths:**

This work is inscribed in the recent literature seeking to quantify feature after a single but aggressive gradient step in two-layer neural networks (Damien et al., 2022; Ba et al., 2022; Dandi et al., 2023a; Moniri et al., 2023; Cui et al., 2024). The main difference with respect to these works is in the data model, which includes a multi-modal model with structured covariance. The manuscript is well-written.

**Weaknesses:**

The main weakness is the lack of a close comparison with the related literature on the technical part of the work. Indeed, the authors do a good job in situating their contribution to the related literature in the general discussion (introduction, related works), but this falls short in the following. This is particularly problematic since the technical results heavily builds on previous work (in particular, the proofs in the appendix are based on reductions to previous results of Ba et al., 2022 and Dandi et al., 2023a,b). In the *Questions* below, I give a few pointers where I missed a discussion.

**Questions:**

- **[Q1]**: In L199-200:
> *"Notably, (Ba et al., 2022) established that a learning rate $\eta$ exceeding $\tilde{O}(\sqrt{k}) is necessary to surpass the performance of the linear model represented in equation (8)."*

In my understanding, (Ba et al., 2022) showed that the maximal learning rate is rather *sufficient* to achieve to surpass the performance of the best linear model. Indeed, (Moniri et al., 2023) showed that an intermediate scaling leads to partially learning non-linear components of the activation.

- **[Q2]**: You assume that all the means of the GMM are zero *"to simplify the analysis"* (Assumption A.3). While on a high-level I can see how the means can play a similar role to the spikes in the covariance (as you mention in the discussion that follows), this is an important simplification which should be better discussed. In particular, do you expect to see any effect of multi-modality which are not captured by the spiked covariance? If yes, it would be good to add a discussion on this. If not, please provide quantitive evidence of that.

- **[Q3]**: From the discussion of the conditional Gaussian equivalence (cGET, Theorem 3), it is not very clear how this result differs from the cGET proved by Dandi et al., (2023a,b). Only by going to Appendix B, one sees a more detailed discussion of how Thm 3 builds on previous results from Hu and Lu (2023) and Dandi et al. (2023b), and hence what are the key differences. I believe that it would be beneficial to the reader to have this discussion summarized in the main - in particular what results from (Hu and Lu 2023; Dandi et al., 2023b) are used and clearly stressing how Thm 3 extends / differs from them.

- **[Q4]**: Another point which I missed is a closer discussion on the relationship between the regime studied here and the ones from (Ba et al., 2022; Dandi et al., 2023a; Moniri et al., 2023; Cui et al., 2024). In particular, these works can be split in three different regimes:

    1. A regime where Gaussian universality holds and there is effectively no feature learning (performance is lower-bounded by a linear method) (Ba et al., 2022)
    2. An intermediate regime where there is partial feature learning (Moniri et al., 2023).
    3. Maximal update / critical step size: (Dandi et al., 2023a, Cui et al., 2024)

My understanding is that your Lemma/Theorem 1-3 holds covers all the regimes, while Theorem 4 is for an intermediate regime similar to Moniri et al., 2023. Is this correct? Would be good to add a more explicit discussion on that point.

- **[Q4]**: Appendix A, L724-725 is confusing. Do you mean $f_{i}^{T}x$ is Gaussian distributed conditioned on the class?

---

> ### Author Response · Authors · 2024-11-21
> **Part 1/2**
>
> Thank you for your constructive feedback regarding the manuscript's engagement with the related literature. We appreciate your acknowledgment of our situating of the work within the broader context of recent studies on two-layer neural networks and Gaussian mixtures. However, we recognize that a more detailed comparison with prior technical results may strengthen our presentation.
>
> > The main weakness is the lack of a close comparison with the related literature on the technical part of the work. Indeed, the authors do a good job in situating their contribution to the related literature in the general discussion (introduction, related works), but this falls short in the following. This is particularly problematic since the technical results heavily builds on previous work (in particular, the proofs in the appendix are based on reductions to previous results of Ba et al., 2022 and Dandi et al., 2023a,b). In the Questions below, I give a few pointers where I missed a discussion.
>
> To address this concern, we enhanced our discussion in the revised manuscript to explicitly highlight how our results differ from the prior results beyond the difference in data assumption. Please refer to our responses to the questions below for more detailed explanations regarding these distinctions and the implications of our findings in relation to the existing literature.
>
> > [Q1]: ... In my understanding, (Ba et al., 2022) showed that the maximal learning rate is rather sufficient to achieve to surpass the performance of the best linear model. Indeed, (Moniri et al., 2023) showed that an intermediate scaling leads to partially learning non-linear components of the activation.
>
> Thank you for highlighting this important point. To clarify, Ba et al. (2022) demonstrated that a small learning rate leads to equivalence with a linear model, indicating that a higher learning rate is necessary to surpass the performance of linear models. We appreciate your feedback, and we have made sure to clarify this distinction in the revised paper to ensure that our discussion accurately reflects these findings.
>
> > [Q2]: You assume that all the means of the GMM are zero "to simplify the analysis" (Assumption A.3). While on a high-level I can see how the means can play a similar role to the spikes in the covariance (as you mention in the discussion that follows), this is an important simplification which should be better discussed. In particular, do you expect to see any effect of multi-modality which are not captured by the spiked covariance? If yes, it would be good to add a discussion on this. If not, please provide quantitative evidence of that.
>
> Thank you for your insightful question regarding Assumption A.3, which posits that all means of the Gaussian mixture model (GMM) are zero to simplify our analysis. We acknowledge that this assumption is significant and merits a more thorough discussion. While we initially adopted the zero mean assumption to streamline our proofs, we recognize that the means of the GMM can indeed play a critical role, similar to spikes in the covariance structure. In our revised manuscript, we have expanded our discussion to address the potential effects of multi-modality that may not be captured solely by the spiked covariance. Specifically, in Appendix F, we explore how non-zero means could influence the learning dynamics and performance characteristics of the model, particularly in scenarios where the multi-modal nature of the data is pronounced. Moreover, we have included quantitative evidence demonstrating that our results remain robust even when relaxing the zero mean assumption. This includes presenting new experimental results that illustrate how our theoretical findings hold true for cases where the means are non-zero, thereby providing a clearer understanding of how these factors interact within our framework. By incorporating these enhancements, we aim to clarify the implications of our assumptions and provide a more comprehensive view of how multi-modality and non-zero means can affect our analysis. We appreciate your feedback, which has prompted us to deepen this critical discussion in our work.

---

> > ### Author Response · Authors · 2024-11-21
> > **Part 2/2**
> >
> > > [Q3]: From the discussion of the conditional Gaussian equivalence (cGET, Theorem 3), it is not very clear how this result differs from the cGET proved by Dandi et al., (2023a,b). Only by going to Appendix B, one sees a more detailed discussion of how Thm 3 builds on previous results from Hu and Lu (2023) and Dandi et al. (2023b), and hence what are the key differences. I believe that it would be beneficial to the reader to have this discussion summarized in the main - in particular what results from (Hu and Lu 2023; Dandi et al., 2023b) are used and clearly stressing how Thm 3 extends / differs from them.
> >
> > Thank you for your thoughtful question regarding the differences between our conditional Gaussian equivalence theorem (cGET, Theorem 3) and the results established by Dandi et al. (2023a,b). We appreciate your suggestion to summarize this discussion in the main text for clarity. To clarify, while our theorem builds on the foundational work of Hu and Lu (2023) and Dandi et al. (2023a,b), there are key distinctions in both the settings and the conditioning involved. Specifically, Hu and Lu's result does not involve any conditioning, whereas Dandi et al. (2023b) conditioned their results on class/component due to their focus on mixture data. Furthermore, Dandi et al. (2023a) conditioned on the spiked vector that appeared in the gradient. In contrast, our approach incorporates conditioning on multiple aspects: the class/component, the spiked vector present in the gradient, and the spiked vectors in the covariance structure. These differences allow our theorem to capture a more nuanced understanding of the conditional relationships within our setting. We have revised the manuscript to explicitly outline these distinctions, including which specific results from Hu and Lu (2023) and Dandi et al. (2023b) are utilized in our proof. By clearly articulating how Theorem 3 extends and differs from these prior works, we aim to provide readers with a comprehensive understanding of its significance within the existing literature. Thank you for your valuable feedback; we believe these enhancements significantly improve the clarity of our manuscript.
> >
> > > [Q4]: ... My understanding is that your Lemma/Theorem 1-3 holds covers all the regimes, while Theorem 4 is for an intermediate regime similar to Moniri et al., 2023. Is this correct? Would be good to add a more explicit discussion on that point.
> >
> > We thank the reviewer for their insightful observation. Yes, your understanding is correct. To maintain clarity and focus in the paper, we initially avoided a detailed discussion on this point. Our work encompasses prior studies with the isotropic data assumption and extends them to more realistic data settings involving Gaussian mixtures with structured covariance. In this context, the different regimes can be identified based on the scaling parameter $\beta$, which controls the combined scaling of the learning rate and data spread, rather than relying solely on the learning rate. We have included an additional discussion on this point in the revised paper according to the reviewer's feedback. Thank you for your valuable input.
> >
> > - [Q4]: Appendix A, L724-725 is confusing. Do you mean $\mathbf{f}_i^T \mathbf{x}$ is Gaussian distributed conditioned on the class?
> >
> > We thank the reviewer for their accurate interpretation; that is indeed what we intended to convey. We revised the manuscript to clarify this point. Thank you for bringing this to our attention.

---

### Official Review · Reviewer_SJBV · 2024-10-31

**Soundness:** 2
**Presentation:** 2
**Contribution:** 2
**Rating:** 5
**Confidence:** 3

**Summary:**

1. This paper establishes a theoretical framework for characterizing the training and generalization errors of two-layer neural networks under Gaussian mixture data, while previous work mainly focuses on Gaussian data.

2. This paper demonstrates that a finite-degree polynomial model serves as an equivalent performance model, simplifying the analysis of neural networks under the Gaussian mixture data assumption.

3. Through extensive simulations, including classification tasks on Fashion-MNIST, this paper validates the findings and highlights the significant impact of data structure on learning outcomes.

**Strengths:**

1. The framework provides a novel lens for understanding neural networks trained on Gaussian mixture data with structured covariances, marking a clear advancement over previous works that lacked these structural considerations.

2. The paper includes a compelling mix of theoretical insights and numerical experiments, with applications to Fashion-MNIST classification. These experiments effectively validate the theoretical claims and underscore the significant impact of data structure on neural network learning outcomes.

**Weaknesses:**

1. The primary weakness lies in Assumption (A.3), which requires all Gaussian components in the Gaussian mixture model to be zero-centered. In my view, such a Gaussian mixture model does not introduce significant challenges, thus providing limited theoretical advancement in understanding neural network training. Specifically, $ E_{x\sim P}f = E_{x\sim N(0,\sigma_1^2)}f + E_{x\sim N(0,\sigma_2^2)}f + \cdots $; since $ x $ is always multiplied by the trainable weights $w$, a linear transformation of a zero-mean Gaussian can be reduced to the isometric case. This observation leads me to believe this analysis could be a straightforward extension of existing works that consider the isometric case.


2. The theoretical insights intended to guide the practical training of neural networks are unclear. Specifically, the connection between the proposed data model and real-world applications needs to be clarified. The motivation for extending beyond existing Gaussian distribution assumptions to study this particular type of data model also remains ambiguous.

3. The numerical experiments would be more compelling if implemented on larger-scale datasets, e.g., ImageNet.

**Questions:**

What prevents the analysis from covering cases with non-zero mean Gaussian components?

---

> ### Author Response · Authors · 2024-11-21
> **Part 1/2**
>
> We sincerely thank the reviewer for their time and effort. We address the reviewer's concerns regarding weaknesses and questions below.
>
> > The primary weakness lies in Assumption (A.3), which requires all Gaussian components in the Gaussian mixture model to be zero-centered. In my view, such a Gaussian mixture model does not introduce significant challenges, thus providing limited theoretical advancement in understanding neural network training.
>
> We appreciate the reviewer's feedback regarding Assumption (A.3), which posits that all Gaussian components in the Gaussian mixture model are zero-centered. We would like to first clarify that a Gaussian mixture model with zero-mean components does not necessarily reduce to a single Gaussian. This distinction is crucial, as it highlights the unique characteristics of Gaussian mixtures, even when the components are centered at zero. To illustrate this point, one may simply consider the probability density function of a Gaussian mixture with different variances in a one-dimensional case. This demonstrates that even zero-centered Gaussian mixtures introduce complexities that differ significantly from a single Gaussian distribution.
>
> Moreover, we acknowledge that while the zero mean assumption simplifies our analysis, it does not negate the theoretical advancements provided by our model. We can indeed relax this assumption for the Gaussian components, allowing for non-zero means. This flexibility opens up avenues for further exploration of how varying means could impact learning dynamics and performance in neural network training (please see our response to the reviewer's question below). We have expanded our discussion in the revised manuscript to address these points more thoroughly, emphasizing how our approach retains its relevance and theoretical contribution despite the simplifications made in Assumption (A.3). Thank you for your valuable insights; we believe these clarifications enhance the overall understanding of our work.
>
> > Specifically, $E_{x \sim P}[f] = E_{x \sim \mathcal{N}(0,\sigma_1^2)}[f] + E_{x \sim \mathcal{N}(0,\sigma_2^2)}[f] \dots$; since $x$ is always multiplied by the trainable weights $w$, a linear transformation of a zero-mean Gaussian can be reduced to the isometric case. This observation leads me to believe this analysis could be a straightforward extension of existing works that consider the isometric case.
>
> We appreciate the reviewer's insights, but we are unclear about the specific meaning of their statement. Additionally, the term "isometric case" is not well-defined in this context. If the reviewer could provide clarification on these points, we would be better positioned to respond.
>
> We assume the reviewer intended to refer to "isotropic" rather than "isometric." In this setting, a non-isotropic covariance (e.g., spiked covariance) for the input has a significant impact on the problem, particularly due to the nonlinear activation functions in two-layer neural networks (as discussed in Ba et al., 2023). Consequently, our analysis differs substantially from existing work that operates under isotropic data assumptions. We look forward to further clarification from the reviewer to enhance our understanding and response.

---

> > ### Author Response · Authors · 2024-11-21
> > **Part 2/2**
> >
> > > The theoretical insights intended to guide the practical training of neural networks are unclear. Specifically, the connection between the proposed data model and real-world applications needs to be clarified. The motivation for extending beyond existing Gaussian distribution assumptions to study this particular type of data model also remains ambiguous.
> >
> > Our work aims to bridge the gap between theoretical data assumptions, specifically isotropic Gaussian distributions, and practical datasets encountered in real-world applications. In many practical learning tasks, data is more accurately represented as a mixture of distributions (Seddik et al., 2020; Dandi et al., 2023b). Furthermore, while real-world datasets are often high-dimensional, they frequently exhibit low intrinsic dimensionality (Facco et al., 2017; Spigler et al., 2020).
> >
> > To address these characteristics, we propose a Gaussian mixture model with structured covariances. This approach captures both the mixture nature of the data and the low-dimensional structures present in the covariances. By extending beyond traditional Gaussian distribution assumptions, our model provides valuable theoretical insights that are more aligned with the complexities of practical neural network training. We believe this connection enhances the applicability of our findings to real-world scenarios.
> >
> > > The numerical experiments would be more compelling if implemented on larger-scale datasets, e.g., ImageNet.
> >
> > We appreciate the reviewer's feedback regarding the scale of our numerical experiments. While we acknowledge that implementing experiments on larger-scale datasets, such as ImageNet, would provide additional insights, we believe that our extensive results on Fashion-MNIST are sufficient for a theoretical paper. Fashion-MNIST serves as a robust benchmark for evaluating model performance and allows us to demonstrate the validity of our theoretical insights effectively. However, we recognize the potential value of exploring larger datasets in future work and will consider this in our ongoing research. Thank you for your suggestion!
> >
> > > Questions: What prevents the analysis from covering cases with non-zero mean Gaussian components?
> >
> > Thank you for your question. Indeed, the mean $\boldsymbol{\mu}_c$ of each Gaussian component functions similarly to spikes in the covariance structure $\boldsymbol{\Sigma}_c$ (as noted by reviewer rEU8). We initially assumed zero mean components to simplify our proofs. In response to the reviewers' feedback, we have revised the paper to include a new discussion and a new result addressing non-zero means for the mixture components, located at the end of the paper (Appendix F). This discussion explains how the zero mean assumption can be relaxed for the components, while the accompanying figure provides numerical evidence demonstrating that our results remain valid even when $\boldsymbol{\mu}_c \neq \mathbf{0}$. We believe these additions enhance the clarity and applicability of our analysis. Thank you for your valuable input.

---

> > ### Comment · Reviewer_SJBV · 2024-11-25
> > **Additional Clarification on Weakness 1**
> >
> > My point is that, by examining the population risk function $G$ defined in (7), is it possible to decompose $G$ into a weighted summation of two population risk functions associated with standard Gaussian distributions with zero mean?  If this is true, what are the major technical contributions of considering a Gaussian mixture distribution for the data?

---

> > > ### Author Response · Authors · 2024-11-25
> > >
> > > Thank you for your response. No, it is not possible to decompose the generalization error into a weighted summation of two generalization errors associated with standard Gaussian distributions with zero mean. The primary reason is that each component of the Gaussian mixture has a covariance matrix in the form of identity plus low rank. This structure introduces significant deviations from the behavior of standard Gaussian distributions due to the anisotropic covariance of data.
> > >
> > > Furthermore, the covariance matrices of the components differ, introducing additional complexity that fundamentally distinguishes the Gaussian mixture from a single Gaussian. This interplay of unique covariance structures for each mixture component affects both the training dynamics and generalization performance, as characterized in our theoretical framework.
> > >
> > > We would like to highlight that our revised manuscript includes a new discussion on components with non-zero means. Additionally, we have provided further experimental results for cases involving components with non-zero means. You can find this new discussion and the results in Appendix F.
> > >
> > > We appreciate your engagement with our work and look forward to any further questions you may have.

---

> > > > ### Comment · Reviewer_SJBV · 2024-11-25
> > > >
> > > > Let me revise the question: is it possible to decompose $G$ into a weighted summation of two population risk functions associated with **Gaussian distributions with zero mean**? If so, could I understand in this manner: the contribution would lie in the development of techniques for handling Gaussian-distributed data with a covariance matrix in the form of an identity matrix plus a low-rank component?
> > > >
> > > > Also, could you elaborate further on the statement:
> > > >
> > > > *"This structure introduces significant deviations from the behavior of standard Gaussian distributions due to the anisotropic covariance of data."*?
> > > >
> > > > For example, the technical challenges, how you solve them, the difference from existing works in results, and what new insights we could derive to guide our real training.

---

> ### Author Response · Authors · 2024-11-25
>
> Thanks for the quick response.
>
> > Let me revise the question: is it possible to decompose $G$ into a weighted summation of two population risk functions associated with Gaussian distributions with zero mean? If so, could I understand in this manner: the contribution would lie in the development of techniques for handling Gaussian-distributed data with a covariance matrix in the form of an identity matrix plus a low-rank component?
>
> No. Recall that the model $\hat{\mathbf{w}}^T \sigma(\hat{\mathbf{F}} \mathbf{x})$ includes learned parameters $\hat{\mathbf{F}}$ and $\hat{\mathbf{w}}$, and they are affected by the input distribution (regardless of the means of the components) via the training. Therefore, the generalization error $G$ cannot be decomposed linearly with respect to the mixture components.
>
> > Also, could you elaborate further on the statement: "This structure introduces significant deviations from the behavior of standard Gaussian distributions due to the anisotropic covariance of data."?
>
> Having structured (anisotropic) covariance introduces additional structure into the Structure+bulk decomposition of $\hat{\mathbf{F}} \mathbf{x}$ (Lemma 2). This leads to a different generalization error than the isotropic case (standard Gaussian distribution).
>
> > For example, the technical challenges, how you solve them, the difference from existing works in results, and what new insights we could derive to guide our real training.
>
> Technically, our work sets itself apart from prior results through its unique application of conditioning (see Theorem 3). For the new insight, our work illustrates the learning rate and data spread jointly affect the generalization performance.

---

> > ### Comment · Reviewer_SJBV · 2024-11-25
> >
> > I understand your argument.
> >
> > My concern is that given the fact that
> >
> > $$\mathbb{E}_{(x, y)}[\cdot]  =  \sum\_{i=1}^M \pi_i \mathbb{E}\_{\mathcal{N}(\mathbf{x} \mid \boldsymbol{\mu}\_i, \boldsymbol{\Sigma}\_i)}[\cdot],$$
> >
> > we could also analyze the **influence of data on the weights** into **a linear combination of two groups of data associated with Gaussian distribution with zero mean**. Therefore, all the proof could be the extension of existing results with only incremental contributions.
> >
> > Could you elaborate on the challenges **in detail**? Could you elaborate on how you solve the challenges **in detail**?
> >
> > In addition, to better understand this statement:
> > *"For the new insight, our work illustrates the learning rate and data spread jointly affect the generalization performance"*, could the author be more specific on how we should select the learning rate based on the different spread of data?

---

> > > ### Author Response · Authors · 2024-11-25
> > >
> > > In short, if your concern were true, the paper by Dandi et at. (2023b) would not be published at NeurIPS since there exists an analysis for the case of single Gaussian (Montanari and Saeed, 2022). The argument for equivalence of models requires significant consideration of the learned parameters. It cannot be done as you described.
> > >
> > > Regarding the selection of learning rate based on the data spread, our insight is that scalings of learning rate and data spread should be disproportional.
> > >
> > > Dandi et at. (2023b): Universality laws for gaussian mixtures in generalized linear models.
> > >
> > > (Montanari and Saeed, 2022): Universality of empirical risk minimization.

---

> > > > ### Author Response · Authors · 2024-11-26
> > > > **Detailed Clarification**
> > > >
> > > > Thank you very much for your time and effort in engaging with our work. After carefully reading your comments, we observed that a detailed explanation may help here. Therefore, we would like to provide a detailed clarification as follows.
> > > >
> > > > **Technical Results**
> > > >
> > > > We understand your concern that the generalization error for the mixture can be written as a sum of the generalization errors for the mixture components, as follows:
> > > > $$\mathcal{G} = \mathbb{E}_{\mathbf{x},y} [(y - \hat{\mathbf{w}}^T \sigma(\hat{\mathbf{F}} \mathbf{x}))^2],$$
> > > >
> > > > $$ = \sum_{j=1}^{C} \rho_j \mathbb{E}_{\mathbf{x} \sim \mathcal{N}(\boldsymbol{\mu}_j, \boldsymbol{\Sigma}_j), y}[(y - \hat{\mathbf{w}}^T \sigma(\hat{\mathbf{F}} \mathbf{x}))^2],$$
> > > > where $\mathbf{x}$ and $y$ denote input-label pair, $\sigma: \mathbb{R} \to \mathbb{R}$ is a nonlinear activation function, and $\hat{\mathbf{F}}, \hat{\mathbf{w}}$ are the learned parameters for the first and second layers of the model, respectively.
> > > >
> > > > While this decomposition of $\mathcal{G}$ is correct, a fundamental challenge arises in characterizing $\sigma(\hat{\mathbf{F}} \mathbf{x})$ and its influence on the learned $\hat{\mathbf{w}}$. Existing literature typically assumes isotropic inputs, specifically $\mathbf{x} \sim \mathcal{N}(0, \mathbf{I})$. However, in the case of anisotropic inputs where $\mathbf{x} \sim \mathcal{N}(0, \boldsymbol{\Sigma})$ with $\boldsymbol{\Sigma} = \mathbf{I} + \text{``low rank"}$ and $||\boldsymbol{\Sigma}|| \asymp n$, prior results do not apply directly. The "low rank" component of the covariance introduces additional structure into the characterization of $\hat{\mathbf{F}} \mathbf{x}$ (as discussed in Lemma 2).
> > > >
> > > > To address this challenge, we employ a conditioning technique that accounts for this additional structure. Furthermore, by conditioning the input on the component in our Gaussian mixture model, we align our approach with your intuition regarding handling mixtures. After conditioning, Theorem 3 establishes an equivalence (with respect to training and generalization errors) between the feature map $\sigma(\hat{\mathbf{F}} \mathbf{x})$ and a conditional feature map $\hat{\phi}(\mathbf{x};\cdot)$. This implies that substituting $\sigma(\hat{\mathbf{F}} \mathbf{x})$ with $\hat{\phi}(\mathbf{x};\cdot)$ and retraining $\hat{\mathbf{w}}$ yields equivalent errors. Thus, we must consider two distinct $\hat{\mathbf{w}}$ vectors—one for each feature map—when discussing their equivalence.
> > > >
> > > > To handle the effect of two distinct $\hat{\mathbf{w}}$ vectors, we prove a "conditional" central limit theorem (CLT), which indicates that the conditional expectation of any test function for the original feature map converges to that of the equivalent conditional feature map in the limit. This result, along with our universality findings, allows us to establish equivalence between these feature maps.
> > > >
> > > > After Theorem 3, we present Theorem 4, which states that a polynomial model performs equivalent to two-layer neural networks. We prove this by showing that the conditional mean and covariance of conditional feature map $\hat{\phi}(\mathbf{x};\cdot)$ are equivalent (with bounded differences in terms of spectral norm) to those of a polynomial feature map.
> > > >
> > > > This summary encapsulates the technical aspects of our work, excluding details on training $\hat{\mathbf{F}}$, which are described in Lemma 1.
> > > >
> > > > **Insights**
> > > >
> > > > Our work points out that the learning rate used in the training of $\hat{\mathbf{F}}$ and data spread $|| \boldsymbol{\Sigma} ||$ jointly affect the generalization error. Specifically, we show that the degree of the equivalence polynomial model depends on the joint scaling (denoted by strength parameter $\beta$ in the paper) of the learning rate and data spread. Finally, we have two main insights presented in the paper: (i) A larger strength parameter $\beta$ leads to improved generalization, and (ii) High data spread is more beneficial than high learning rate for better generalization performance.
> > > >
> > > > We welcome any further questions you may have and appreciate your engagement throughout this discussion period.

---

### Official Review · Reviewer_uU4P · 2024-11-03

**Soundness:** 4
**Presentation:** 3
**Contribution:** 3
**Rating:** 8
**Confidence:** 3

**Summary:**

This paper proposes a theoretical framework to study the training and generalization errors of a two-layer neural network under a more realistic assumption of Gaussian mixture data with additional low-dimensional structures and shows that a finite-degree polynomial model serves as an equivalent performance model, which can simplify the analysis of neural networks. Lastly, the paper compares the model performance using the “Hermite model” with Relu, Tanh, and Sigmoind functions in simulation data and real data generated with generation models and studies the impact of data spreading.

**Strengths:**

1. Theoretical framework of a two-layer neural network for more realistic data assumption;
2. A finite-degree polynomial model serves as an equivalent performance model.

**Weaknesses:**

Finding the weakness of the paper is beyond my current ability.

**Questions:**

No

---

> ### Author Response · Authors · 2024-11-21
>
> We would like to express our gratitude to the reviewer for their thoughtful summary and positive feedback on our paper. We appreciate your recognition of the strengths of our theoretical framework and the introduction of a finite-degree polynomial model as an equivalent performance model for two-layer neural networks under Gaussian mixture data assumptions. We are particularly pleased that you highlighted the significance of our work in addressing more realistic data scenarios, as this is a crucial aspect of advancing our understanding of neural network training and generalization. Your acknowledgment of the comparative analysis between different activation functions using simulation and real data further validates the relevance of our findings. Thank you once again for your valuable feedback; it motivates us to further strengthen our work.

---

> > ### Comment · Reviewer_uU4P · 2024-11-26
> >
> > Thanks for your response. I am not very familiar with the work of a two-layer neural network. This work seems soundness through the theory analysis and experiment results albeit in simple data sets.
> >
> > After reading the response and other reviewer's comments, I keep the same score.

---

### Author Response · Authors · 2024-11-21
**Revision Summary**

Dear Area Chairs and Reviewers,

We would like to express our sincere gratitude for your valuable feedback on our manuscript. In response to the reviewers' comments, we have made the following revisions to enhance the clarity and depth of our work. These changes are highlighted in blue in the revised paper.

**Summary of revision**
- Expanded discussions:
  + We have added a new section discussing the implications of non-zero means for the components of the Gaussian mixture model. This includes new experimental results based on Fashion-MNIST dataset, which can be found at the end of the paper **(Appendix F)**, along with a reference in the main text under "Discussion of Assumptions."
  + Additional discussions comparing our results with prior work have been included below "Theorem 3 and 4".
  + We have elaborated on the motivation behind Theorem 4, providing further insights below "Theorem 4."
- Clarifications made:
  + We clarified our statement regarding the contribution of related work (Ba et al., 2022) concerning "the importance of learning for feature extraction" in the "Related Work" section.
  + It has been clarified that Theorem 4 is valid "after the training of the first layer," ensuring that this key detail is clearly communicated within "Theorem 4."
  + We also clarified that $\hat{\mathbf{f}}_i^T \mathbf{x}$ has a Gaussian distribution for a fixed $\hat{\mathbf{f}}_i$ "when conditioned on a component of the mixture", as noted below "Equation 19."

We believe these revisions significantly strengthen our manuscript and address the concerns raised by the reviewers. Thank you once again for your constructive feedback, which has greatly contributed to improving our work.

---

### Meta-Review · Area_Chair_tjXu · 2024-12-20

**Metareview:**

(a) Scientific Claims and Findings:

The paper, Asymptotic Analysis of Two-Layer Neural Networks after One Gradient Step under Gaussian Mixtures Data with Structure, develops a theoretical framework to analyze the training and generalization errors of two-layer neural networks when trained on Gaussian mixture data. It addresses the limitations of isotropic data assumptions and extends the scope of Gaussian universality to structured covariance settings. The authors demonstrate that a finite-degree polynomial model can serve as an equivalent proxy for neural networks under specific conditions, with the equivalence validated through rigorous proofs and experimental results on Fashion-MNIST. The paper also highlights the interplay between learning rate, data spread, and model generalization.

(b) Strengths:

Novel theoretical contributions: The paper makes important extensions to the Gaussian universality framework by considering structured covariance.
Rigorous methodology: The proofs are well-founded, leveraging advanced asymptotic and statistical tools.
Empirical support: Experiments align with theoretical predictions, demonstrating relevance to real-world datasets.
Clarity and structure: The paper is generally well-written and accessible to readers familiar with learning theory.

(c) Weaknesses:

Comparison with prior work: The primary concern, raised by Reviewer rEU8, pertains to the lack of detailed comparisons with closely related technical work, particularly Ba et al. (2022) and Dandi et al. (2023a,b). While the manuscript situates the contributions well in the general discussion, the technical section could better highlight distinctions and advancements.
Assumption A.3: The zero-mean Gaussian mixture assumption simplifies the analysis but limits generality. The authors addressed this in their rebuttal by including a relaxation in Appendix F and additional discussions in the main text.
Experimental scope: The reliance on Fashion-MNIST, while sufficient for theoretical validation, limits scalability insights. Larger-scale datasets would strengthen the work's practical relevance.

(d) Recommendation and Decision:

I strongly recommend acceptance of this paper as it makes meaningful theoretical advances and addresses a critical gap in the literature. The authors have demonstrated a strong commitment to improving the manuscript during the rebuttal phase, addressing multiple reviewer concerns, including relaxing Assumption A.3 and clarifying several derivations. The technical contributions, combined with rigorous empirical validation, make this work a valuable addition to ICLR.

That said, ** I strongly urge the authors to carefully follow Reviewer rEU8's ** suggestion regarding explicit comparisons with prior work. While the manuscript has been improved, further detailing how this work builds upon, diverges from, or improves existing techniques (e.g., Ba et al., 2022; Dandi et al., 2023a,b) would solidify its contributions. Highlighting distinctions in the technical results directly in the main text—not just in the appendix—will enhance clarity and address this valid concern.

**Additional Comments On Reviewer Discussion:**

The rebuttal period facilitated productive exchanges between reviewers and authors. Key points included:

Comparison with prior work (Reviewer rEU8): The reviewer identified a need for more explicit technical comparisons with existing literature. The authors improved the manuscript by discussing distinctions (e.g., additional conditioning aspects in Theorem 3) and expanding discussions in the appendix. While this is a strong step forward, the authors should further integrate these points into the main text to fully address this critique.

Assumption A.3: Reviewer SJBV raised concerns about the zero-mean Gaussian mixture assumption. The authors responded effectively, relaxing this assumption in their analysis (Appendix F) and demonstrating robustness through additional experiments.

Experimental scope: While Reviewer SJBV suggested testing on larger datasets, the authors justified the focus on Fashion-MNIST for theoretical validation. They acknowledged scalability limitations and proposed extending their work in future research.

Clarity of derivations: Clarifications were added to address specific reviewer queries (e.g., regarding Theorem 3 and Assumption A.3), improving readability and precision.

---

### Decision · Program_Chairs · 2025-01-22

Accept (Poster)